# Ice particle sampling from aircraft – influence of the probing position on the ice water content

Armin Afchine[1], Christian Rolf[1], Anja Costa[1], Nicole Spelten[1], Martin Riese[1], Bernhard Buchholz[2], Volker Ebert[2], Romy Heller[3], Stefan Kaufmann[3], Andreas Minikin[3], Christiane Voigt[3], Martin Zöger[3], Jessica Smith[4], Paul Lawson[5], Alexey Lykov[6], Sergey Khaykin[7], and Martina Krämer[1]

[1]Forschungszentrum Jülich, Institute of Energy and Climate Research (IEK-7), Jülich, Germany;
[2]Physikalisch-Technische Bundesanstalt Braunschweig, Germany
[3]Deutsches Zentrum fr Luft- und Raumfahrt, Institute for Physics of the Atmosphere, Oberpfaffenhofen, Germany
[4]Harvard University, Cambridge, MA, USA
[5]SPEC Incorporated, Boulder, CO, USA
[6]Central Aerological Observatory (CAO), Department of Upper Atmospheric Layers Physics, Moscow, Russia
[7]LATMOS/IPSL, UVSQ Université Paris-Saclay, UPMC University Paris 06, CNRS, Guyancourt, France

*Correspondence to:* Martina Krämer (m.kraemer@fz-juelich.de)

**Abstract.** The ice water content (IWC) of cirrus clouds is an essential parameter determining their radiative properties and thus is important for climate simulations. Therefore, for a reliable measurement of IWC on board of research aircraft, it is important to carefully design the ice crystal sampling and measuring devices. During the ML-CIRRUS field campaign in 2014 with the German Gulfstream GV HALO (High Altitude LOng range) aircraft, IWC was recorded by three closed path total water together with one gas phase water instrument. The hygrometers were supplied by inlets mounted on the roof of the aircraft fuselage. Simultaneously, the IWC is determined by a cloud particle spectrometer attached under an aircraft wing. Two more examples of simultaneous IWC measurements by hygrometers and cloud spectrometers are presented, but the inlets of the hygrometers were mounted at the fuselage side (M-55 Geophysica, StratoClim campaign 2017) and bottom (NASA WB57, MacPex campaign 2011). This combination of instruments and inlet positions provides the opportunity to experimentally study the influence of the ice particle sampling position on the IWC with the approach of comparative measurements. As expected from theory and shown by Computational Fluid Dynamics (CFD) calculations, we found that the IWCs provided by the roof inlets deviate from those measured under the aircraft wing. Caused by the inlet position in the shadow-zone behind the aircraft cockpit, ice particles populations with mean mass sizes larger than about 25 μm radius are subject to losses, which lead to strongly underestimated IWCs. On the other hand, cloud populations with mean mass sizes smaller than about 12 μm are dominated by particle enrichment and thus overestimated IWCs. In the range of mean mass sizes between 12 and 25μm, both enrichment and losses of ice crystals can occur, depending on whether the ice crystal mass peak of the - in these cases bimodal - size distribution is on the smaller or larger mass mode. The resulting deviations of the IWC reach factors of up to 10 or even more for losses as well as for enrichment. Since the mean mass size of ice crystals increases with temperature, losses are more pronounced at higher temperatures while at lower temperatures IWC is more affected by enrichment. In contrast, in the cases where the hygrometer inlets were mounted at the fuselage side or bottom, the agreement of IWCs is -due to less disturbed ice particle sampling, as expected from theory- most frequently within a factor of 2.5 or better, independently of the

mean ice crystal sizes. The rather large scatter between IWC measurements reflects for example cirrus cloud inhomogeneities, instrument uncertainties as well as slight sampling biases which might occur also at the side or bottom of the fuselage and under the wing. However, this scatter is in the range of other studies and represent the current best possible IWC recording on fast flying aircraft.

## 1 Introduction

Cirrus ice water content (IWC) is directly linked to the clouds extinction and thus relates bulk cloud properties to radiative properties (e.g. Gayet et al., 2004; Heymsfield et al., 2014; Thornberry et al., 2017). Since IWC is a parameter representing cirrus in global climate models, a solid knowledge of IWC is of importance. The most accurate measurements are achieved by
in-situ aircraft observations where cirrus clouds are directly probed. However, the measurements must be carried out carefully to obtain the desired data quality. Beside the ability of the instruments that are used to detect the complete range of IWCs with sufficient accuracy, the probing position at the aircraft's fuselage is of importance (see Krämer et al., 2013, and references therein).

The IWC is a bulk quantity which is composed of the sum of all masses of ice particles of different sizes contained in an air
volume. Yet there are shadow and enrichment zones for ice crystals, which depend on the ice particle size and and the position relative to the fuselage. These zones are the most prominent particle measurement bias caused by an aircraft body. Thus, in case the position for particle sampling is placed in such a zone, it can be expected that an IWC measurement will be distorted. These effects are described already by airflow and trajectory calculations in King (1984) for different sized cloud particles. In particular, King (1984) shows that above the roof of an aircraft the sampling of particles is greatly disturbed. However, to
simulate and quantify losses or enrichment of ice particles and the effect on particularly the IWC at a specific position of an aircraft is hardly possible, since this depends on the prevailing ice particle size distribution and flight conditions.

Here, we use a comparative experimental approach to determine the influence of particle probing positions on IWC measurements of cirrus clouds, by relating in-situ observations of IWC measured at the roof, side, bottom and under the wing of aircraft with different instruments. Specifically, IWC is measured under the wing - which is the most favorable position for
particle sampling - during three field campaigns with differing aircraft. One aircraft is additionally equipped with three other IWC instruments placed at the aircraft roof, at the second the IWC measurement is placed at the aircraft side and at the third at the aircraft bottom. From the comparison of the correlation of the roof, side and bottom to the wing IWCs conclusions are drawn about the representativeness of the measurements at the specific position. The results of the measurements at the aircraft roof are validated by exemplary CFD simulations of gas streamlines and ice particle trajectories around the aircraft for typical conditions during penetrations of cirrus clouds.

## 2 Methodology

To determine the quality of an IWC measurement performed on airplanes is challenging, because the IWC evolves from a population of ice crystals of varying size that can be influenced by flow perturbations caused by the aircraft. In a perfect system, all ice particles of each size that are contained in a volume of undisturbed air would be collected. However, even small distortions of the airflow in comparison to calm air conditions can cause deviations in the IWC. These and other effects that depend on the size of the crystals can distort the IWC measurement in different ways and it is difficult to reproduce their influence on IWC.

To understand the effects that may occur for specific ice particle sizes, CFD simulations of gas streamlines and particle trajectories around an airplane are helpful. These effects can be caused for example by unfavorable sampling positions together with specific flight conditions such as the aircraft speed and the planes angle of attack. For specific cases, potential shadow or enrichment zones can be identified and the effect on IWC can be estimated. These estimates, however, differ for each particle size and, in addition, the particle concentration of each size must be known to determine the overall influence on the IWC. This influence can also vary for each IWC measurement with the ice particle size distribution ($PSD_{ice}$), flight conditions and related changes of the shadow and enrichment zones.

On the other hand, all effects that may occur as a result of flow disturbances or other causes (discussed in the last paragraph of this section) are included in the measurement of the bulk IWC. Hence, for the evaluation of the quality of IWC measurements an experimental comparative approach of IWC measurements is useful. The explanatory power of comparative IWC measurements is described in the following. The first step of the approach is to establish a reference bulk IWC measurement with respect to the instrument performance (i.e. good precision of the measurement). This is achieved by gas phase and total water measurements with different instruments mounted on a fuselage (see Sections 4.1 and Figure 9).

Next, the bulk IWC is compared to an IWC measurement at a differing position, here at the aircraft wing, which is least susceptible to flow disturbances if it is properly positioned (see Section 4.2). In this study, the wing IWC is derived from the measurements of $PSD_{ice}$ (see also Section 4.2), that should be only weakly influenced by flow perturbation effects. An agreement of the wing IWC with the bulk IWC measured on the fuselage (shown in Section 5.1.3) could indicate that both measurements are influenced in the same way by flow perturbations or instrument and other effects - but this seems not very likely because of the very different flow conditions for the sampling positions under the wing and on the roof. We interpret such an agreement in the way that both measurements are little influenced by airflow or instrument and other effects. Such a reliable agreement between IWCs from two different instruments mounted at two different positions is a reasonable indication for an applicable IWC measurement. Vice versa, as soon as the ice particle sampling at one or both positions is seriously disturbed by effects outlined in the next paragraph, the IWC measurements will differ significantly from each other (see Section 5.1.2). As will be shown in Section 5.2, from such IWC deviations it is possible to draw conclusions on the manner of the IWC distortion, for example if the probing position is placed in a shadow or enrichment zone. Also, the IWC deviations from each other can be quantified by using the comparative IWC approach.

However, some scatter between IWCs measured with different instruments and at different positions must be expected. The reasons for this are manifold: first of all, cirrus clouds are very inhomogeneous, even on the scale of the distance of the measuring instruments from each other, so the differing probe mounting positions can cause differing IWCs. Also, each mounting position on a fast flying aircraft, even when chosen as careful as possible, might be slightly influenced by distortions of the airstream in comparison to the calm air conditions and thus can cause deviations in IWCs. Further, bouncing ice crystals

may break and the small fragments may enter the IWC sampling areas and, also, the density of air can influence the particle sizes that enter these areas. Last, some unknown uncertainties are always included in the derivation of IWC from the $PSD_{ice}$. For example, the applied parametrizations are derived from measurements with a certain scatter, and, the particle counting statistics can be poor in thin cirrus clouds. The resulting overall scatter between IWC measured in this study is shown in Section 5.1.4.

## 3   IWC measurements - a brief excursion into theory

As introduced in the previous section, the IWC of cirrus can be recorded from aircraft either by bulk cloud measurements using airborne closed path hygrometers mounted behind an inlet tube or via integration of the ice particle number size distributions, $PSD_{ice}$, measured by cloud spectrometers. In both cases, the ice particles must be properly sampled before the measurement. The bulk IWC is less error-prone in comparison to the IWC from $PSD_{ice}$ in case of undisturbed ice particle sampling. The

reason is that before the bulk measurements the ice crystals are evaporated while the size resolved IWC detection must account for the ice crystal shapes. In the following, a brief summary on sampling and measuring IWC on fast flying aircraft is given. For more detail, we refer to e.g. Krämer and Afchine (2004), Schiller et al. (2008), Wendisch and Brenguier (2013), Krämer et al. (2013), Luebke et al. (2013).

### 3.1   IWC from hygrometers

The bulk IWC is derived from the difference between $H_2O_{tot}$, which is the amount of total water ($H_2O_{gas}$ + evaporated ice crystals) contained in a cirrus, and $H_2O_{gas}$, the gas phase water amount. The IWC is calculated by using the following Equation:

$$IWC = H_2O_{tot} - H_2O_{gas} = \frac{H_2O_{enh} - H_2O_{gas}}{E_{max}} \tag{1}$$

where $H_2O_{enh}$ ($H_2O_{tot}$ enhanced by an oversampling of ice crystals) and $E_{max}$ (enhancement factor) are parameters related to

the sampling of the ice crystals by an inlet tube which is described in Section 3.1.2.

For the measurement of $H_2O_{gas}$, the air loaden with water vapor is passed into the aircraft by an inlet tube which faces against the direction of flight. Therefore, a pump is used to suck the air through the inlet-hygrometer-exhaust line. No cloud particles enter backward facing inlets, since their inertia is too high for a complete U-turn. The hygrometer is mounted behind the inlet in the aircraft cabin.

To measure $H_2O_{tot}$ (or $H_2O_{enh}$, respectively) is more difficult, since also ice particles of a wide range of sizes ($\approx 3 - 1000$ µm or

more in cirrus clouds) has to be passed into the aircraft. To this end, inlet tubes facing into the direction of flight are deployed. To precisely determine $H_2O_{tot}$, the ice crystals have to be completely evaporated before they enter the hygrometer, which is placed subsequently in the sampling line. For that, the inlet should be heated to up to 90°C. In addition, a strong bend should

follow directly behind the inlet to shatter ice crystals to small fragments that evaporate in a short time. Behind the water measurement the air leaves the aircraft at the outlet point. Most systems are so-called 'free stream' sampling lines, i.e. the flow is generated by the pressure difference between the inlet tip and the outlet. Prerequisite for a reliable $H_2O_{tot}$ measurement is a suitable, well-characterized inlet so that the true concentration of water plus evaporated ice crystals can be determined. To accomplish this requirements, two points are important: (i) First, the inlet needs to be placed at the aircraft fuselage in a way to

enable sampling in undisturbed flow. (ii) Further, the inlet itself should minimally influence the gas phase water and ice particle concentration. These two points are briefly described in the following, mainly based on Krämer et al. (2013) and references therein.

### 3.1.1 IWC enrichment or loss due to inlet position

The principle behavior of gas streamlines and cloud particle trajectories around an aircraft fuselage can be seen in Figure 1

(adapted from King, 1984). In the upper panel of these early, but still meaningful potential flow simulations, the predicted gas flow streamlines at 90 m/s are displayed. Far in front of the aircraft's nose they are equally spaced, indicating the same flow velocity. However, due to the aircraft body the streamlines are compressed over the cockpit, indicating regions of higher airspeed -and also enriched concentrations of smaller cloud particles that follow the streamlines- compared to the free stream.

In the bottom panel, trajectories for larger (exemplarily 100 µm) cloud particles are displayed for the same flight conditions.

As these particles have high inertia, most of the trajectories end at the aircraft fuselage, i.e., the particles impact on the aircraft. However, some of the trajectories were deviated, leading to regions devoid of particles (shadow zone) or with increased particle concentration (enrichment zone).

To specify the aforementioned size ranges of the 'smaller' and 'larger' cloud particles, CFD calculations for the specific conditions of fuselage shape, aircraft speed and inlet distance from the nose of the aircraft need to be performed. Very roughly,

cloud particles with radii <30 µm can be assumed to belong to the smaller, while those >30 µm are associated to the larger part of the cloud particle size spectrum at jet aircraft with high air speeds. Altogether, when measuring cloud particles on the roof of an airplanes, it is important to know where shadow and enrichment zones on the aircraft platform are located, since at the same fuselage station it is possible to sample in the shadow/enrichment zone for larger/smaller particles if a probe is positioned close to the aircraft fuselage or in the enrichment zone for larger particles in case the probe is farther away from the fuselage.

To minimize the effect of streamline compression and deviation of particle trajectories during the sampling of cloud particles, it is favorable to mount the sampling inlets on the aircraft's side or bottom well apart of the fuselage. There, the flow is much closer to free stream conditions, and the largest deviations from these conditions occur near the fuselage and in regions of strong curvature (Twohy and Rogers, 1993). Most favorable for an undisturbed sampling on aircraft is most likely the position under an aircraft wing with the probes head ahead of the aircraft wing, since the aerodynamically shaped wing has the least

influence on the flow.

### 3.1.2 IWC enhancement due to inlet design

The first requirements to an inlet for a proper sampling are that it protrudes beyond the aircraft's boundary layer and that the wall of the inlet tip is thin enough to avoid strong shattering of ice crystals or deviation of streamlines from the free flow. However, as explained in the following, a deviation from the gas streamlines is desirable when sampling cirrus clouds, since cirrus are very thin and their IWC can be as low as $10^{-3}$ ppmv ($\sim 10^{-4}$ mg/m$^3$). To this end, so called 'nearly virtual impactors' (see Figure 2) are used for the collection of cirrus ice particles. These are inlets where the velocity inside of the inlet tube (U) is much smaller than the flow speed ($U_0$). Actually U is so small ($U/U_0 < 0.2$, e.g. Krämer and Afchine, 2004)) that the inlet cross section appears like an impaction plate. Such inlets sample strongly 'sub-isokinetic', i.e. the part of the cross section where gas streamlines enter the inlet is much smaller than the part of the cross section that samples ice particles. The particle sampling cross sections increases with increasing particle size up to the total inlet cross section for the largest particles. As a consequence, ice crystals are sampled from a much larger (enhanced) air volume than $H_2O_{gas}$ and thus the combined sampling of $H_2O_{gas}$ and evaporated ice crystals is also enhanced ($H_2O_{enh}$ instead of $H_2O_{tot}$). To adjust the two volumes to each other, the ice crystal air volume (and thus the IWC, see Eq. 1) needs to be corrected for this enhancement.

As mentioned, the enhancement (which can also be called 'aspiration efficiency') is dependent on particle size and increases for larger particles, up to a maximum value $E_{max}$. This maximum value is used for the calculation of the IWC (see Eq. 1). $E_{max}$ can be calculated from the velocity of the free stream $U_0$ and the velocity U inside of the inlet:

$$E_{max} = \frac{U_0}{U} \tag{2}$$

The point where the enhancement is 50% of $E_{max}$ ($E_{50}$) is called the 'cut-off' size of the inlet which defines the particle size range sampled by the inlet. $E_{max}$ is dependent on U, which in turn depends, among other parameters like pressure, temperature and aircraft speed $U_0$, strongly on the pressure difference between inlet and outlet, the driving force of the flow (in case the flow rate is not controlled). Thus, U decreases with increasing altitude.

With the knowledge of $E_{max}$, the IWC can now be calculated following Eq. 1. In Figure 3, we visualize the complex relation between the measuring parameter $H_2O_{enh}$, IWC and $E_{max}$ in dependence of temperature for given $H_2O_{gas}$ (assumed as the saturation value for the calculations), calculated from Eq. 1 (left column: $E_{max} = 10$, right column: $E_{max} = 50$; top row: volume mixing ratio, bottom row: concentration). To avoid very small artificial IWCs caused by the uncertainties of measurements and not by ice particles, the minimum difference between $H_2O_{enh}$ and $H_2O_{gas}$ needs to be 5% to encounter an IWC. The differently colored regions show the ranges of $H_2O_{enh}$ and IWC belonging to each another. It can be seen from Figure 3, that the IWCs covered by $H_2O_{enh}$ of the same color are broader and show lower IWCs at higher temperatures and narrower with higher IWCs at lower temperatures. This reflects the fact that $H_2O_{gas}$ decreases with temperature and is thus stronger enhanced due to the addition of ice crystals. Consequently, $H_2O_{enh}$ 'jumps' to a higher value with another color. Because of this, the IWC detection

limit as well as the uncertainty of IWC improves with decreasing temperature. Regarding the difference between $E_{max} = 10$ and 50 (left and right panels of Figure 3) it becomes visible that the higher $E_{max}$, the smaller the IWC that can be detected.

The range of IWCs that can be detected with a $H_2O_{tot}$ instrument can be seen from Figure 3. The blue $H_2O_{enh}$ isolines through the IWC-T parameter space correspond to the detection limit of an instrument, e.g. the '1ppmv' and '3ppmv' $H_2O_{enh}$ isolines represent the IWC detection limit of the FISH and HAI instruments that will be described in Section 4.1.2. Further, the IWC detection range is limited at the lower end of IWC in dependence of temperature by the requirement that $H_2O_{enh}/H_2O_{gas}$ > 1.05. A difference of 5% between the two measurements is necessary to avoid that artificial clouds emerge caused by the scatter of the instruments (see also Schiller et al., 2008).

## 3.2   IWC from cloud spectrometers

Cloud spectrometers measure the cloud particle number size distribution $PSD_{ice}$. They are in most cases mounted below the the aircraft wings with distances ahead of the wing and from the aircraft body to minimize particle losses or enrichment due to distorted cloud particle trajectories contamination by cloud particles bounced from the air frame (Krämer et al., 2013). In any case, deviations of streamlines does not play a great role in the flow around wings for particle measurements. To avoid uncertainties in the measurements caused by the aircraft's angle of attack, the cloud probes should be mounted under this angle to compensate this effect. Ice crystal shattering into small fragments ($\sim$<50 μm diameter) at the cloud probes head is a source of error in $PSD_{ice}$. However, this does not play a significant role for the calculation of the IWC – for cloud probes equipped with anti-shattering inlet tips – since the ice fragments contribute to the integrated mass of $PSD_{ice}$ in the same way as the as the original crystal several hundred microns or more in size. For those cloud spectrometers that use anti-shattering tips and data evaluation algorithms, ice fragments from large shattered ice crystals can be considered (Korolev et al., 2011). However, without these tools ice crystals from outside could shatter at the inlet tips and the small fragments are then being swept into the sample volume. Other measurement issues of $PSD_{ice}$ are discussed in detail in (Krämer et al., 2013) and Baumgardner et al. (2017).

The IWC is derived from $PSD_{ice}$ by summing up the ice crystal concentrations measured in each size bin of the number size distribution. The largest source of error in this method is the irregularity of the ice crystal shapes. Especially large ice crystals cannot be assumed as spheres and their shapes strongly vary. Numerous so-called mass-dimension (m–D) or mass-area (m-A) relations are derived to account for this effect (a comparison is shown in Section 4.2). A summary of m-D relations is given e.g. in Abel et al. (2014) and a new, advanced relation is developed by Erfani and Mitchell (2016). The m–D relations are of the form:

$$m_i = a \cdot D_i^b \tag{3}$$

with $m_i, D_i$ mass and diameter of the ice crystals of the i-th size bin and a, b constants of respective relations. The IWC is then:

$$IWC = \sum_{i=1}^{n} m_i \cdot dN_i \tag{4}$$

# 4 IWC instrumentation

## 4.1 Bulk IWC inlet and hygrometers

### 4.1.1 $H_2O_{tot}$ inlets

For the HALO aircraft, Trace Gas Inlets (TGI) are designed[1], mainly to probe atmospheric gas components, but also to sample ice cloud particles. The design can be seen in the bottom panel of Figure 4, where a TGI is mounted with three inlets facing in forward direction for cloud sampling and one inlet in backward direction for gas constituents. The height of the TGI and the distances of the inlets from the fuselage are designed to protrude from the aircraft's boundary layer, the numbers are listed in Table 1. The TGI inlet is heated, and the sampling tubes have a 90° bend as required to evaporate ice crystals entering the forward facing ducts (see Section 3.1.) During ML-CIRRUS in 2014, two TGIs were mounted on the frontmost apertures of HALO's roof. The roof position was chosen for the various apertures due to technical restrictions. Two $H_2O_{tot}$ hygrometers (FISH and Waran, for description of the $H_2O$ instruments see next section) are positioned at the upper forward inlet tips of TGI 1 and 2, a third hygrometer (HAI) is connected to the middle forward duct of the TGI 1. The TGI position at the aircraft fuselage is shown in the top panel of Figure 4. The hygrometer used for $H_2O_{gas}$ sampling (SHARC) is connected to a backward inlet tip of a TGI mounted further downstream.

On board of Geophysica, the inlet for the $H_2O_{tot}$ hygrometer FISH is mounted at the side of the aircraft, as can be seen in Figure 5. It is also heated and has a 90° bend. The $H_2O_{gas}$ hygrometer FLASH is mounted below a wing and equipped with it's own inlet. The WB-57 $H_2O_{tot}$ inlet for the FISH hygrometer is mounted at the aircraft's bottom (see Figure 6) and is as well heated and has a 90° bend. The $H_2O_{gas}$ hygrometer HWV is mounted below a wing and equipped with it's own inlet. The IWCs derived from the $H_2O_{tot}$ measurements behind the respective inlets are here referred to as roof, side and bottom IWCs.

### 4.1.2 $H_2O$ instruments

The essentials of the hygrometers used to measure $H_2O_{tot}$ and $H_2O_{gas}$ on board of HALO during ML-CIRRUS 2014 (FISH, HAI, Waran and SHARC) are summarized in the following. For more detail we refer to the respective cited publications of the instruments.

FISH (Fast In situ Stratospheric Hygrometer) is a closed path Lyman-$\alpha$ photofragment fluorescence (Zöger et al., 1999; Meyer et al., 2015) to measure $H_2O_{tot}$ in the range of 1- 1000 ppmv between 50-500 hPa with an accuracy/precision of 6–8%/0.3 ppmv. Connected to the HALO-TGI forward facing duct, the enhancement factor range is 12-20. In accordance to Figure 3, the resulting minimum detectable IWC is between about $1\text{-}20 \cdot 10^{-3}$ ppmv ($\sim 1\text{-}20 \cdot 10^{-4}$ mg/m$^3$). The time resolution of the measurements is 1 Hz.

HAI (Hygrometer for Atmospheric Investigation) is a four channel Tunable Diode Laser hygrometer (Buchholz et al., 2017). Here, we use its closed path 1.4 μm $H_2O_{tot}$ channel, for brevity called HAI in the following. The measurement range is 3 - 40000 ppmv with an accuracy/precision of 4.3%$\pm$3 ppmv/0.24 ppmv. Its enhancement factor at the HALO-TGI is 17-50,

---

[1] *enviscope* GmbH.

the resulting minimum IWC following Figure 3 is between about $0.5\text{-}20 \cdot 10^{-2}$ ppmv ($\sim 0.5\text{-}20 \cdot 10^{-3}$ mg/m$^3$) and the time resolution is 1 Hz.

Waran (Water Vapor Analyzer) is a tunable diode laser hygrometer (1.4 µm) WVSS (Vance et al., 2015), attached to the forward facing TGI (Voigt et al., 2017) instead of the originally associated inlet. The detection range is $\gtrsim$50–40000 ppmv, the accuracy according to the manufacturer is $\pm$50 ppmv or 5% of reading, whatever is larger. However, good performance of

WVSS down to about 20 ppmv is reported in Smit et al. (2013) in a comparison of airborne hygrometers. The enhancement factor at the HALO-TGI is in the range of 20-35 and the resulting minimum detectable IWC is (see Figure 3) between about $0.5\text{-}50 \cdot 10^{-1}$ ppmv ($0.5\text{-}50 \cdot 10^{-2}$ mg/m$^3$) at a time resolution of 0.4 Hz.

SHARC (Sophisticated Hygrometer for Atmospheric Research) is also a closed path Tunable Diode Laser hygrometer (1.4 µm), but at HALO used for $H_2O_{gas}$ measurements (Meyer et al., 2015). Its range of detection is 20-40000 ppmv with an accu-

racy/precision of 2-4%/0.2 ppmv at a time resolution of 1 Hz.

On board of Geophysica during StratoClim 2017, $H_2O_{tot}$ was measured by FISH, while for $H_2O_{gas}$ FLASH (FLuorescent Airborne Stratospheric Hygrometer, for details see Khaykin et al., 2013) was used. FLASH uses also the Lyman-$\alpha$ photofragment fluorescence technique for the detection of water vapor, but its inlet is designed to sample only the gas phase. The

detetction range is 1-1000 ppmv with an accuracy/precision of <9%/0.5 ppmv, the time resolution is 1 Hz.

FISH was also used for $H_2O_{tot}$ measurements on board of the WB-57 during MacPex 2011. In this case, $H_2O_{gas}$ is detected by the Lyman-$\alpha$ fluorescence hygrometer HWV (Harvard Water Vapor, time resolution of 1 Hz). Details of the water measurements during MacPex are described in Rollins et al. (2014).

## 4.2 Cloud spectrometers for IWC

During ML-CIRRUS 2014 and also StratoClim 2017, the NIXE-CAPS (New Ice eXpEriment: Cloud and Aerosol Particle Spectrometer, NIXE hereafter) instrument, mounted under the wing of HALO (see Figure 7) and Geophysica, respectively, was used to measure the cloud particle number size distribution in the size range of 3-930 µm diameter at a time resolution of 1 Hz (Meyer, 2012). The mounting positions (distance from leading edge of the wing and distance to wing surface) are

listed in Table 1. Comprehensive CFD studies had been performed during the modification of the plane to a research aircraft to determine the optimal position for particle sampling (but without permission to be shown). Two instruments are incorporated in NIXE: the NIXE-CAS-DPOL (Cloud and Aerosol Spectrometer with Detection of POLarization) and the NIXE-CIPg (Cloud Imaging Probe - Greyscale). In combination, particles with diameters between 0.61 µm and 937 µm can be sized and counted. For cloud measurements, particle diameters $> 3$ µm are considered. The data analysis methods and all applied correction algorithms are described in Meyer (2012) and Luebke et al. (2016). The IWC was derived using the m-D relation described by Krämer et al. (2016) and Luebke et al. (2016). This relation, originally derived from observations by Mitchell et al. (2010) and confirmed in the study of Erfani and Mitchell (2016), has nearly no dependency on temperature or cirrus type, thus demonstrating the robustness of the connection between cirrus ice crystal size and mass. The m-D relation is again confirmed

by our measurements, which can be seen by the good agreement of IWCs derived from PSDs from NIXE-CAPS with those determined from total water measurements with FISH (see Figure 11, left panel). Furthermore, it should be noted that the IWCs derived from PSDs are not very sensitive on the choice of the m-D relation. That can be seen in Figure 6, where, in addition to the above mentioned m-D relations, the also usual m-D relations of Heymsfield et al. (2010) and Cotton et al. (2013) are plotted in the left panel. The right panel of Figure 6 shows IWCs calculated from the 10 different m-D relations versus their mean IWC for one flight during ML-CIRRUS. It can be seen that the IWCs from the m-D relations are at most around the factor 1.5 over the entire IWC range. Specifically, 55% of the data range between 1:±1.2, while 19/26% can be found in the ranges 1:-(1.2 to 1.5) / 1:(1.2-1.5).

During MacPex 2011, the cloud spectrometer 2D-S (Lawson et al., 2006) was mounted under a wingpod of the WB-57 to measure cloud particles at a time resolution of 1 Hz (the mounting position is listed in Table 1). 2D-S is an optical imaging cloud probe comparable to the CIPg, covering the particle size range of 15-1280 μm diameter. The IWC is derived from an a-D (area-dimension) relation described by Baker and Lawson (2006) which is again confirmed here (see Section 5.1.3 and Figure 11, right panel).

The IWCs derived from the wing mounted NIXE or 2D-S ice particle measurements are here referred to as 'Wing IWCs'.

# 5 Ice particle probing position and IWC

## 5.1 IWCs from roof/side/bottom and wing sampling

### 5.1.1 Roof $H_2O$ measurements

First, the measurements of the hygrometers mounted at roof of the HALO aircraft (FISH, HAI, Waran and SHARC) are compared to each other to ensure that possible instrument differences are not attributed to the probing position in the further discussion. Note here that the FISH instrument is a well-established hygrometer with a long history of successful aircraft measurements and instrument intercomparisons (Fahey et al., 2014; Rollins et al., 2014; Meyer et al., 2015). SHARC, HAI and Waran are developed for and first deployed on the HALO aircraft.

To this end, scatter plots of $H_2O$ in clear air as well as IWCs in cirrus are shown in Figure 9. Good agreement of the clear air $H_2O$ measurements (at $RH_{ice} < 60\%$ to strictly exclude clouds) from FISH, HAI and SHARC is demonstrated in the left panel of the figure. The middle panel show the IWC scatter plot of FISH and HAI. Most of the measurements symmetrically spread around the 1:1 line within a factor of 2.5, which can be considered as a good agreement (as discussed in Section 2 and Section 5.1.4). Linear regression is calculated for the data range > 0.2 ppmv, representing the lower detetction limit of HAI in the observed temperature range (see Figure 3). The correlation coefficient $R^2 = 0.82$ (the correlation coefficients are given in the figure caption, the regression lines are not plotted to keep the visual clearness of the graphics). In the right panel, the measurements of FISH and Waran are displayed. The data are mostly placed above the 1:1 line, most frequently around a factor of 2.5. This means that the IWC of Waran is shifted to higher values in comparison to FISH. An explanation for this behavior

is still missing. The linear regression is calculated for the data range $> 0.5$ ppmv, the lower detection limit of WARAN in the observed temperature range (see Figure 3), the correlation coefficient $R^2 = 0.89$.

### 5.1.2 Roof and wing IWCs

IWCs from measurements at the aircraft roof in comparison to the IWC measured under the wing are shown in Figure 10. The left/middle/right panels of the figure depict roof-mounted FISH/HAI/WARAN versus wing-mounted NIXE observations.

The first to note is the relatively broad scatter of all IWC measurements. This can be seen from the broad distribution of the data points between the black dashed lines in the panels, which represent a factor of $\pm10$ to the black solid 1:1 line. A closer look to the panels by taking notice of the frequencies of occurrence (see color code in the figure), however, shows

narrower structures parallel to the 1:1 lines. For the FISH instrument, at medium IWCs most data pairs are placed between the 1:1 and 1:2.5 lines (IWC enrichment), while at higher IWCs the highest frequencies are found below the 1:10 line (IWC losses). The same is found for HAI, but at medium IWC losses are seen more often than for FISH. Vice versa, for Waran an IWC enrichment is more abundant and expands beyond the 1:2.5 line in the medium IWC range. No clear correlations can be observed here, as expected when sampling ice crystals on the roof of an airplane where the measurement is influenced by

shadow/enrichment zones for larger/smaller particles (see Section 3.1.1). The structures of IWC deviations seen in Figure 10 will be further analyzed in Section 5.2). What can already be seen when comparing the scattering of IWCs with that around the 1:1 line of the NIXE cloud spectrometer IWCS (see Figure 6, IWCs from different m-D relations), is that the m-D relation is not the cause for the deviations seen in Figure 10.

### 5.1.3 Side/bottom and wing IWCs

To investigate if the differences of the IWCs from roof and wing measurements found in the last section might be indeed related to the $H_2O_{tot}$ inlet position at the aircraft's roof, we analyze IWCs correlations of side/wing and bottom/wing measurements in the following.

Side IWCs were measured by FISH ($H_2O_{tot}$, see inlet position in Figure 5 and Table 1) together with the hygrometer FLASH

for $H_2O_{gas}$, while wing IWCs are recorded by the cloud spectrometer NIXE during the recent field campaign StratoClim 2017 (http://www.stratoclim.org/) with the Russian aircraft Geophysica. Note here that the roof and wing ice particle measurements are performed with instruments also operated on board of HALO. Under clear sky conditions the hygrometers agree as well as those shown in Figure 9, left panel (not shown here).

A good agreement of side/wing IWCs can be seen from the left panel of Figure 11. The majority of data pairs distribute here

between the thin lines, representing a factor of $\pm2.5$.

Linear regression is calculated for the data range $> 0.15$ ppmv, the lower IWC detection limit of NIXE. The correlation coefficient $R^2 = 0.90$ is the highest of the considered correlations. Since for these measurements the same instruments as for the roof/wing measurements were used for ice particle sampling, the position of the $H_2O_{tot}$ inlet at the side of the aircraft is most probably the cause for the better agreement of the IWCs in comparison to the roof/wing IWCs discussed in the previous

section (shown in Figure 10). The reason is that here the airflow clings better at the aircraft fuselage because the cockpit is less disturbing. Consequently, the trajectories of the ice crystals are not deflected, as it occurs at the roof of the aircraft (see Section 3). Another aspect of the good agreement between the two measurements is that it shows the validity of the m-D relation used to calculate the IWC from the $PSD_{ice}$ measured by NIXE.

Bottom and wing IWCs were measured by FISH for $H_2O_{tot}$ (see inlet position in Figure 6; note that FISH is also deployed at HALO and Geophysica) and the hygrometer HWV for $H_2O_{gas}$, complemented by the cloud spectrometer 2D-S. The instruments are mounted on the US aircraft WB-57 during the field campaign MacPex 2011 (see Krämer et al., 2016). FISH and HWV agreed well under clear sky conditions (not shown here).

It can be seen from Figure 11, right panel, that - beside that mostly high IWCs are found in the probed mesoscale convective cloud systems - the bottom/wing data pairs are also evenly distributed between the 1:1 and 1:±2.5 lines as for the side/wing observations. This is again attributed to the position of the $H_2O_{tot}$ inlet at the bottom of the aircraft where the ice crystals are not deflected. The correlation coefficient $R^2 = 0.83$ of the linear regression (note that no lower instrument detection limits need to be considered here since the IWCs are generally high) is slightly less than for the side/wing measurements at Geophysica.

### 5.1.4 Scatter of IWC measurements

In all cases of reasonable agreement between IWC measurements, in the sense of the possible agreement between IWC measurements from different ice particle sampling positions discussed in Section 2, the IWC data distributes around the 1:1 line mostly in between a factor of ±2.5 or better (see Figure 9: roof/roof, and Figure 11: side/wing and bottom/wing), represented by the thin lines in the Figures. This is in good agreement with a study of de Reus et al. (2009), where IWCs from $H_2O_{tot}$ (FISH and FLASH) and cloud spectrometers (FSSP and CIP) measurements at the Russian aircraft Geophysica are compared during the field campaign SCOUT-$O_3$. de Reus et al. (2009) reported an IWC scatter of ±2.2 around the 1:1 line. A scatter of IWC data in this order of magnitude is also reported by Thornberry et al. (2017), who measured IWCs by means of the side mounted NOAA-TDL hygrometer and the wing mounted cloud spectrometers FCDP and 2D-S on board of the Global Hawk during the ATTREX 2014 campaign. Abel et al. (2014) reported this quite large scatter, which in all cases exceeds the uncertainties stated for the instruments. The scatter of IWC from three instruments mounted on the WB-57 reported by Davis et al. (2007) is slightly better.

### 5.2 Impact of ice crystal size on roof IWC

To further investigate the structures seen in the roof/wing IWC scatter plots discussed in Section 5.1.2 (see Figure 10), we analyze the influence of the ice particle size distribution ($PSD_{ice}$) on the IWCs and also ice particle trajectories of different sizes around the planes fuselage for the specific case of roof sampling considered here.

To visualize the influence of $PSD_{ice}$ on IWC, we look at the ratio of the roof to the wing IWCs in dependence of the mean mass radius $R_{ice}$ of the $PSD_{ice}$ ($R_{ice} = \left( \frac{3 \cdot IWC}{4\pi\rho \cdot N_{ice}} \right)^{1/3}$ with $\rho = 0.92$ g/cm$^3$) from NIXE, $N_{ice}$ = total number of ice crystals

with diameter $> 3\,\mu m$). The results are shown in Figure 12. In case of undisturbed sampling at both positions at the aircraft, the distribution of the data points should be homogeneous around the 1-line of the IWC-ratio, with the highest frequencies closest to this line. However, the data distribution are more 'duck' shaped for all three roof-mounted $H_2O_{tot}$ instruments. The appearance of the IWC ratios can be divided in three regimes, marked by the thin vertical red lines in Figure 12.

(1) An IWC 'enrichment regime' is observed for small $R_{ice}$ (about $< 12\,\mu m$). A mass size distribution typical for this regime is displayed in Figure 13 ($PSD_{ice}$ 1, top panel; note that for the portrayal of the PSDs we here use the ice particle diameter and not radius to clearly distinguish from the mean mass radius $R_{ice}$ of the ice particle population used in Figures 12). The ice mass of $PSD_{ice}$ 1 accumulates at smaller sizes, larger ice particles does not contribute to the IWC. Following Section 3 (Figure 1), smaller ice crystals at the aircraft roof are enriched close to the fuselage and this is what Figure 12 shows, in consistency with the enrichment at lower IWC seen in Figure 10.

Supportive to this finding based on the experimental approach of comparative IWC measurements, we performed three-dimensional CFD calculations of gas streamlines and ice particle trajectories around an aircraft with a HALO-type fuselage, shown in Figure 14. In panel (a) trajectories of ice crystals as small as $5\,\mu m$ diameter are plotted (thick lines). It can be seen from the Figure that the gas streamlines (thin lines, color coded by the velocity of the flow) are compressed, in accordance with the potential flow calculations. Consequently, the probed air volume is compressed for smaller ice crystals which follow the streamlines, which leads to the observed enrichment of IWC.

(2) An IWC 'loss regime' is detected in Figure 12 for large mean mass $R_{ice}$ (about $\gtrsim 25\,\mu m$). Here, the IWC originates mainly from large ice crystals connected to $PSD_{ice}$ 3 in Figure 13 that are not sampled in the shadow zone at the aircraft roof. A shadow zone can also be seen in the CFD simulation in Figure 14, panels (b) and (c). Ice particles of 50 and $100\,\mu m$ miss the inlet or hit the plane, respectively. Note that the width of the shadow zone differ for the different particle sizes, it increases for the $100\,\mu m$ ice particles in comparison to those with $50\,\mu m$. However, also some cases of IWC oversampling (IWC ratios $> 1$ in Figures 12) are found for large ice crystals. This might be explained by cases where huge ice crystals are present, which meet the inlet directly, as can be seen in panel (d) of Figure 14 ($500\,\mu m$ particle trajectory), but come from air outside of the original sampling volume.

(3) An IWC 'even-handed regime' is found (Figure 12) for intermediate $R_{ice}$ (about 12-25 $\mu m$). The corresponding typical $PSD_{ice}$ 2 can be seen in the middle panel of Figure 13. This type of $PSD_{ice}$ is bimodal with one ice mass peak at smaller and another at larger sizes. Depending on which of the peaks is dominating, the accumulation of smaller ice crystals in the aircraft's enrichment zone or the losses of larger ice crystals in the shadow zone overbalance.

The 'duck' shape of the IWC ratios of the three instruments slightly differ from each other. Most equally distributed around the ratio 1 are the FISH/NIXE IWCs (top panel of Figure 12), with the highest frequencies in the enrichment part of the 'even-handed regime' at IWC ratios slightly above 1. HAI/NIXE IWC ratios (middle panel of Figure 12) on the other hand have the highest frequencies in the loss part of the 'even-handed regime' reaching IWC ratios significantly below 1. This is consistent with the fact that the HAI instrument is connected to the middle forward inlet (see Figure 4) and is thus -in comparison to the FISH inlet- closer to the fuselage. Here, the losses of large particles are more pronounced. Notable is that already a few

5 centimeter have such an effect on the particle sampling efficiency. The bottom panel of Figure 12 shows the Waran/NIXE IWC ratios. Waran is connected -as FISH- to the roof inlet of a TGI right next to that of FISH and thus shows a comparable distribution of frequencies, but shifted to higher values. This reflects that the Waran IWCs are in general somewhat higher than those of the other instruments (see Figures 9 and 10).

## 5.3 Roof and wing IWC climatologies

10 An overview of the impact of the sampling position on the IWC is given in Figures 15, where IWC frequencies of occurrence are shown in dependence of temperature for the roof-mounted FISH instrument (top panel) and the wing-mounted NIXE (bottom panel).

Comparing the roof and wing IWCs at warmer temperatures, it can be clearly seen that high IWCs are not measured at the roof position and thus the higher frequencies are shifted to lower IWCs. The reason is most probably that high IWCs at 15 temperatures $\gtrsim$ 220 K are related to large ice crystal sizes belonging to the 'loss regime' discussed in the previous section, which can be seen in Figures 15 (bottom panel), where frequencies of occurrence of $R_{ice}$ in dependence of temperature are plotted. At lower temperatures, the mean mass ice crystal sizes $R_{ice}$ shrinks into the 'even-handed' and 'enrichment regime' that means they are often enriched, resulting in an overestimation of the roof IWCs. This can be seen in the higher frequencies of larger roof IWCS in comparison to the wing IWCs.

20 Altogether, the IWC climatology of the roof IWCs covers roughly the same range as that of the wing IWCs, with the exception that large IWCs at high temperatures are missed. However, the distribution of the frequencies of occurrence of the IWCs is, caused by the position of the $H_2O_{tot}$ inlet, heavily skewed for the roof IWCs.

## 6 Summary and conclusions

The influence of the ice particle sampling position on IWC measurements on aircraft is investiagted with the approach of 25 comparative measurements. The reproducibility of the underlying total water measurements is assessed by comparison with several instruments at the same position as well as with as phase water instruments. The representativeness of the corresponding IWC measurements at roof, side and bottom mountings on the fuselage is evaluated by comparison with IWCs derived from ice particle size distributions measured under the aircraft wing.

The side and bottom IWC in comparison to wing IWC measurements show a reasonable good agreement. Most frequently 30 they correspond to each other within a factor of 2.5, independently of the mean ice crystal sizes. The reason for the only little disturbed measurements at these positions is that under the aircraft wing and at the side and bottom of the fuselage, the cloud particle trajectories are not greatly diverted caused by the aircraft body or the wing itself, so that the sampling of ice crystals represent nearly ambient conditions. But, the agreement of the IWCs does not only show the performance of the side, bottom and wing sampling position, but also the credibility of the measurements. This is notable since the measurement techniques greatly differ, the side/bottom IWC is measured by the Lyman–$\alpha$ fluorescence hygrometer FISH and the wing IWC is obtained from the ice particle mass size distribution measured by optical methods with NIXE-CAPS and 2D-S. A further conclusion

from the agreement of the IWCs is that it demonstrates the validity of the m-D relation of Erfani and Mitchell (2016), slightly modified by Krämer et al. (2016) and Luebke et al. (2016), which is applied to convert the NIXE-CAPS size of the ice crystals

into mass. In addition, a comparison of ten different m-D relations shows that the resulting IWCs differ from their mean IWC by at most a factor of 1.5 (55% of the data range between 1:±1.2) over the entire IWC range.

However, roof and wing IWCs differ from each other. Since from the side and bottom in comparison to wing measurements the instrument performance is shown, we attribute the differences to the mounting position on the roof. Deviations of the streamlines and particle trajectories above the roof due to the cockpit can lead to both, enrichment and losses of particles

depending on the size of the ice particles. Large ice particles are lost in the shadow-zone behind the aircraft's cockpit, while at the same time smaller ice crystals are enriched. These -expected- findings from the approach of comparative measurements are supported by CFD simulations performed for different ice particle sizes. A more detailed analysis shows that for the measuremets performed in this study the mean mass radii of the ice particle population smaller than about 12 μm, enrichment of the ice crystals and thus an overestimation of the IWC dominates. In the size range 12 to about 25 μm both enrichment

and losses of ice crystal occurs, while loss of large crystals leading to strongly underestimated IWCs prevails for larger sizes. Enrichment and losses are in the order of a factor of 10 or more.

A correction of the IWCs measured at aircraft roofs might only be possible when ice particle PSDs are measured simultaneously. However, in that case the IWCs calculated from the PSDs would still be more accurate. Because of the high variability of the ice particle size distributions, it is also not an option to assume PSDs, e.g. in dependence of temperature, for a correction

of the roof IWCs.

The influence of the size dependent enrichment or losses of ice crystals from the roof sampling propagates to IWC climatologies with respect to temperatures. At higher temperatures, where the ice crystals are larger, IWCs are underestimated due to the ice particle losses, while at lower temperatures overestimation of IWC caused by particle enrichment dominates.

The recommendations resulting from this comparison of in-situ measurements of IWC are that (i) reliable measurements of

IWC are possible from sampling positions at the side, bottom and under the wing when using (ii) instruments with a detection range that cover the complete wide IWC range from about 0.001 to 3000 ppmv, and (iii) placing the instruments far enough away from the fuselage to minimize possible effects of flow distortions. The best approach to measure IWC is to deploy a combination of two instruments at different sampling positions. As last remark we like to note that this recommendations also applies to other ice particle measurements, such as ice crystal numbers sampled by counterflow virtual impactors (Mertes et al.,

30   2007).

*Author contributions.*

A. Afchine: NIXE-CAPS measurements and IWC analysis; M. Krämer: FISH and NIXE-CAPS measurements, IWC analysis; C. Rolf: FISH measurements and IWC analysis; A. Costa: NIXE-CAPS measurements; N. Spelten: FISH measurements; M. Riese: FISH and NIXE deployment; B. Buchholz: HAI measurements; V. Ebert: HAI measurements; R. Heller: Waran measurements; S. Kaufmann: Waran measurements; C. Voigt: Waran measurements; M. Zöger: SHARC measurements; P. Lawson:

2D-S measurements; J. Smith: HWV measurements; A. Lykov: FLASH measurements; S. Khaykin: FLASH measurements ; A. Minikin: under wing cloud spectrometer configuration.

*Acknowledgements.* The authors would like to thank the DFG (Deutsche Forschungsgemeinschaft, German Research Foundation) Priority Program SPP 1294 for funding of the FZJ project 'ACIS' (KR 2957/1-1) and the PTB HAI project (EB 235/3-1 and EB 235/3-2). C. Voigt thanks for finding by HGF contract No. W2/W3-60 and by DFG SPP HALO 1294 contract No. V01504/4-1. Special thanks to our colleagues S. Mertes (TROPOS, Leipzig) and H. Ziereis (DLR, Wessling) for stimulating and important discussions on the topic of ice crystal sampling from aircraft roof.

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

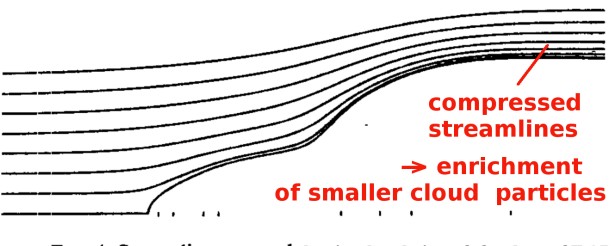

FIG. 4. Streamlines around the simulated aircraft fusalage of F-27.

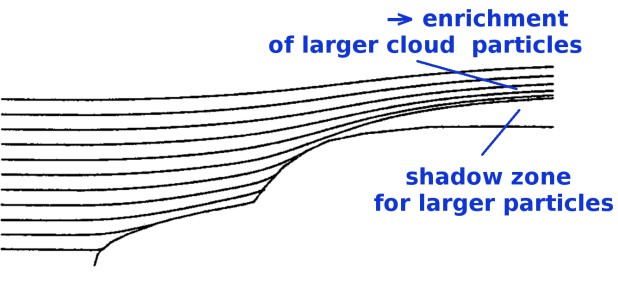

Trajectories around the F-27 for water drops of diameter
100 $\mu$m travelling at 90 m s$^{-1}$, $\eta = 1.75 \times 10^{-5}$ kg s$^{-1}$.

**Figure 1.** Three-dimensional potential flow simulations of gas streamlines and particle trajectories around an aircraft shaped body, adapted from King (1984) (with annotations).

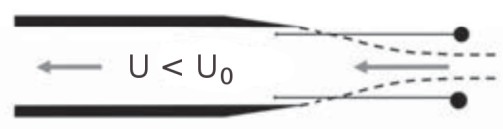

**Figure 2.** Sub-isokinetic sampling of ice particles by a nearly virtual inlet, where the velocity U inside of the inlet tube is much smaller than the flow speed $U_0$. The dashed lines denote the region of the free stream from where the gas streamlines enter the inlet; the black dots illustrate large particles that do not follow the gas streamlines, particle tracks are indicated by thin solid lines (adapted from Krämer et al., 2013).

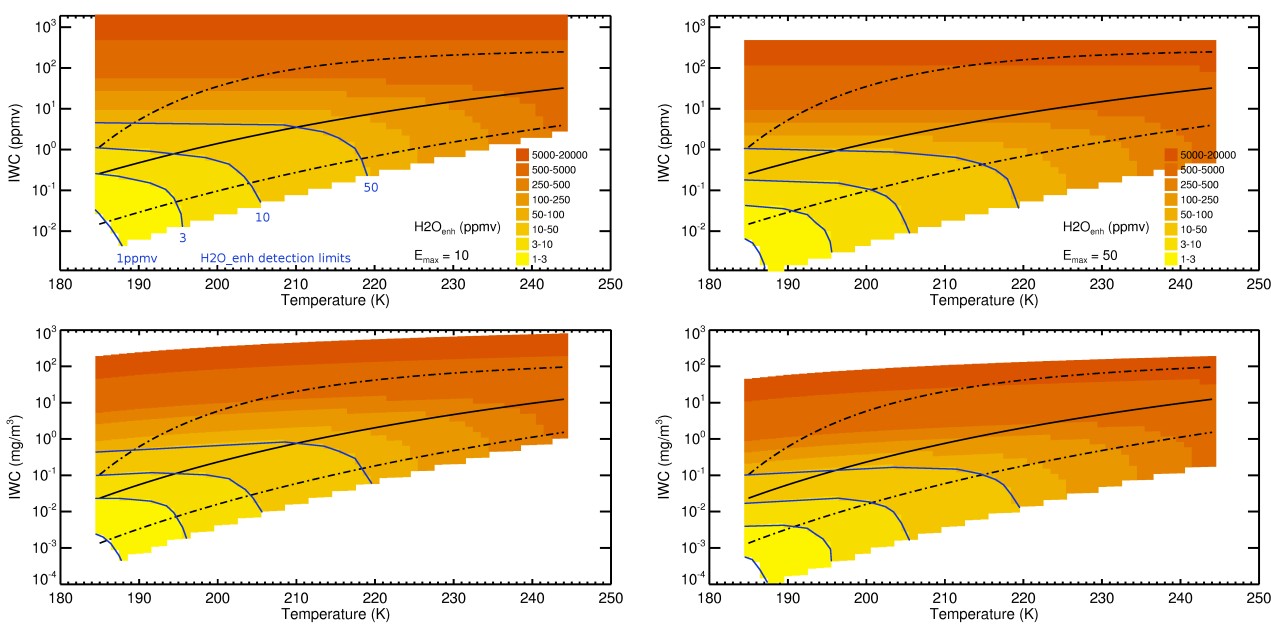

**Figure 3.** Relation between $H_2O_{enh}$ and IWC in dependence of temperature for given $H_2O_{gas}$ (assumed as water vapor saturation value), calculated from Eq. 1 (IWC $= \frac{H_2O_{enh} - H_2O_{gas}}{E_{max}}$) for two different $E_{max}$ (left: 10, right: 50); top: volume mixing ratio, bottom: concentration). The minimum difference between $H_2O_{enh}$ and $H_2O_{gas}$ to detect IWC is 5% to account for measurement uncertainties, i.e. in the white region below the calculated IWCs, $H_2O_{enh}/H_2O_{gas} < 1.05$. Blue lines: $H_2O_{enh}$ isolines corresponding to the detection limit of an instrument, the '1ppmv', '3ppmv' and '10ppmv' $H_2O_{enh}$ isolines represent the IWC detection limit of the FISH, HAI and WARAN instruments described in Section 4.1.2. Black solid and dashed lines: medium, core max and min IWCs after Schiller et al. (2008).

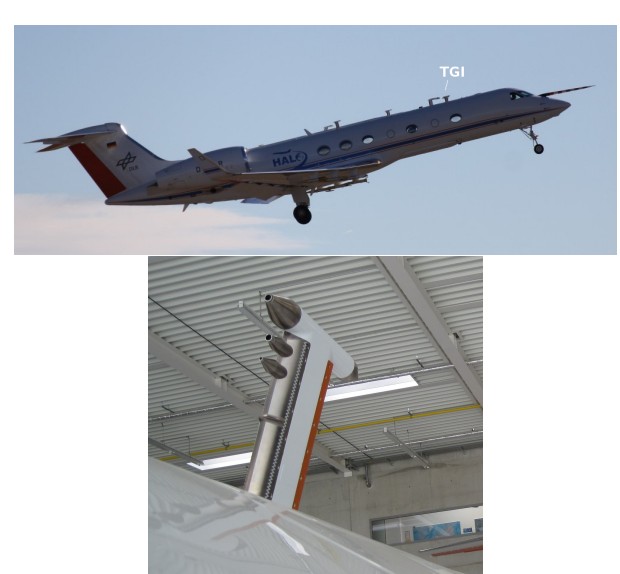

**Figure 4.** Roof mounted FISH, HAI, Waran inlet at HALO (photos: top A. Fix, bottom A. Afchine).

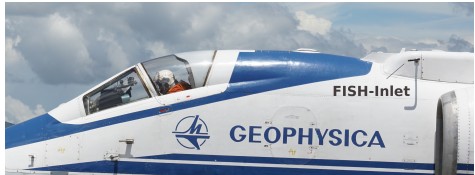

**Figure 5.** Side mounted FISH-inlet at Geophysica (photo: A. Afchine).

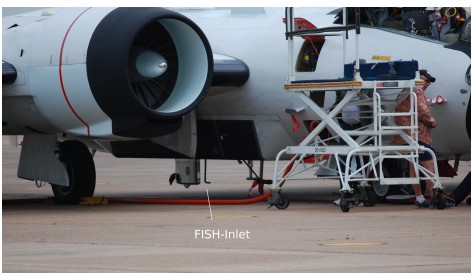

**Figure 6.** Bottom mounted FISH-inlet at WB-57 (photo: A. Afchine).

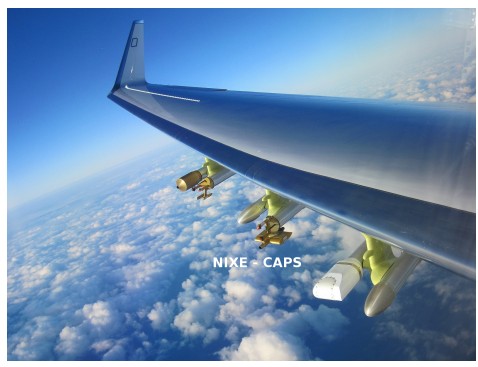

**Figure 7.** NIXE underwing mounting at HALO (photo: A. Afchine).

| HALO (Gulfstream GV) | | | | | | | |
|---|---|---|---|---|---|---|---|
| **Inlet** (roof) | | | | | | **PMS** (wing) | |
| **FISH** | | **HAI** | | **Waran** | | **NIXE-CAPS** | |
| Distance from nose [cm] | Distance to fuselage [cm] | Distance from nose [cm] | Distance to fuselage [cm] | Distance from nose [cm] | Distance to fuselage [cm] | Distance from leading edge of the wing [cm] | Distance to wing surface [cm] |
| 650 | 31.9 | 650 | 26.4 | 650 | 31.9 | 15 | 30 |

| Geophysica (M-55) | | | | WB-57 (NASA) | | | |
|---|---|---|---|---|---|---|---|
| **Inlet** (side) | | **PMS** (wing) | | **Inlet** (bottom) | | **PMS** (wing) | |
| **FISH** | | **NIXE-CAPS** | | **FISH** | | **2-DS** | |
| Distance from nose [cm] | Distance to fuselage [cm] | Distance from leading edge of the wing [cm] | Distance to wing surface [cm] | Distance from nose [cm] | Distance to fuselage [cm] | Distance from leading edge of the wingpod [cm] | Distance to wingpod surface [cm] |
| 500 | 35 | 50 | 50 | 900 | 35 | - 150 | 50 |

**Table 1.** Positions of the total water inlets and cloud spectrometers at the three aircraft (see Figures 4 - 7).

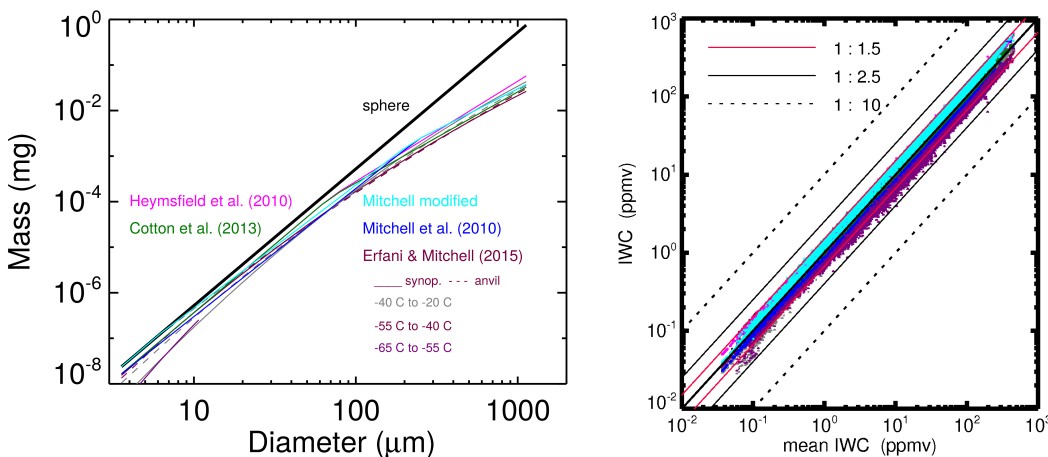

**Figure 8.** Left: Ice particle mass in dependence on size: summary of mass-dimension (m-D) relations. The black line is for an ice sphere. Right: IWCs calculated from the different m-D relations vs. their mean IWC for one flight during ML-CIRRUS (March 29, 2014) where the entire IWC range is covered by the measurements. 55% of the data range between 1:±1.2, while 19/26% can be found in the ranges 1:-(1.2 to 1.5) / 1:(1.2-1.5).

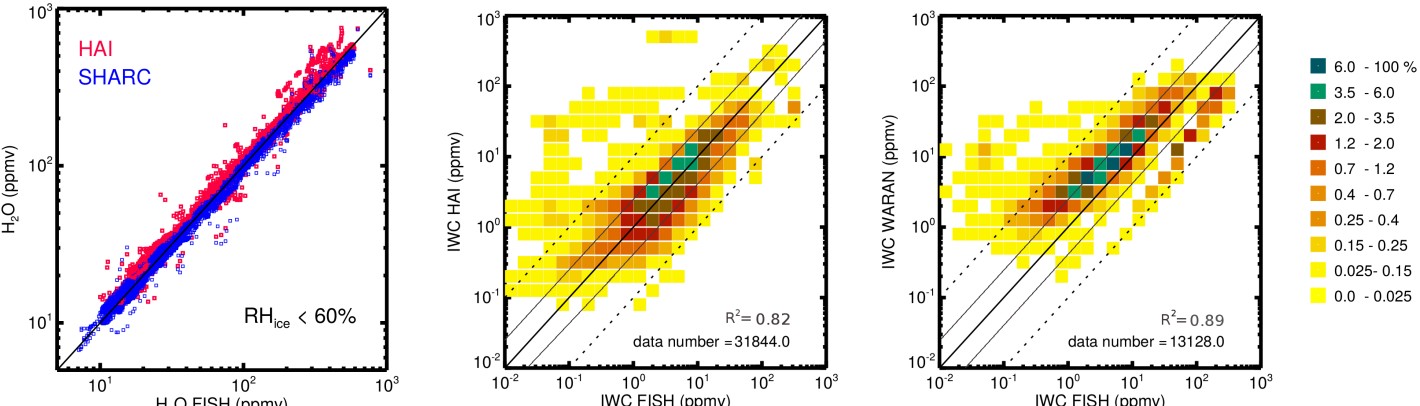

**Figure 9.** Comparison of $H_2O$ and IWCs from roof-mounted closed-path hygrometers FISH, HAI and WARAN ($H_2O_{tot}$) and SHARC ($H_2O_{gas}$) @HALO during ML-CIRRUS 2014 (color code: frequencies; solid black: 1:1 line; dashed/thin: $\pm$ factor 10/2.5 to 1:1 line). Linear regression coefficients for X=IWC FISH, Y=IWC HAI/WARAN are: FISH/HAI (middle) Y = 0.781·X + 0.119, sigma = 0.0032; FISH/WARAN (right) Y = 0.761·X + 0.472, sigma = 0.0035; the coerrelation coefficients $R^2$ are shown in the respective panels. The FISH/HAI regression is calculated for the data range > 0.2 ppmv (lower detetction limit of HAI in the observed temperature range, see Figure 3) and the FISH/WARAN regression for > 0.5 ppmv (lower detection limit of WARAN, see Figure 3)

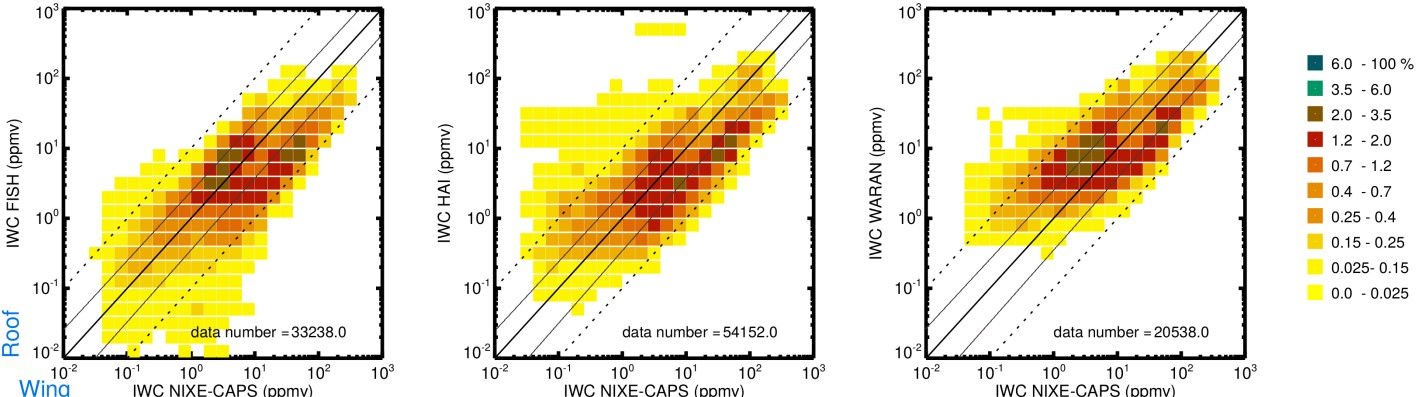

**Figure 10.** Comparison of IWCs from roof-mounted closed-path hygrometers FISH, HAI and Waran (see Equation 1, $H_2O_{gas}$ from SHARC) and wing-mounted cloud spectrometer NIXE @HALO during ML-CIRRUS 2014 (color code: frequencies; solid black: 1:1 line; dashed/thin: ± factor 10/2.5 to 1:1 line).

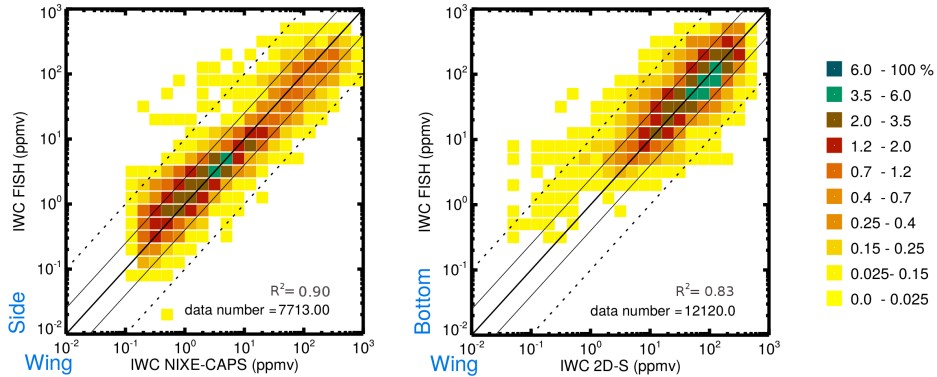

**Figure 11.** Comparison of IWCs from left: side mounted closed-path FISH (see Equation 1, $H_2O_{gas}$ from FLASH) and wing-mounted cloud spectrometer NIXE @Geophysica during StratoClim 2017; right: bottom-mounted closed-path FISH (see Equation 1, $H_2O_{gas}$ from HWV) and wing-mounted cloud spectrometer 2D-S @WB-57 during MacPex 2011 (color code: frequencies; solid black: 1:1 line; dashed/thin: ± factor 10/2.5 to 1:1 line). Linear regression coefficients for X=IWC Wing, Y=IWC Side/Bottom are: Side/Wing (left) Y = 0.768·X + 0.066, sigma = 0.0045; Bottom/Wing (right) Y = 0.856·X + 0.174, sigma = 0.0048; the coerrelation coefficients $R^2$ are shown in the respective panels. The Side/Wing regression is calculated for the data range > 0.15 ppmv (lower detetction limit NIXE-CAPS).

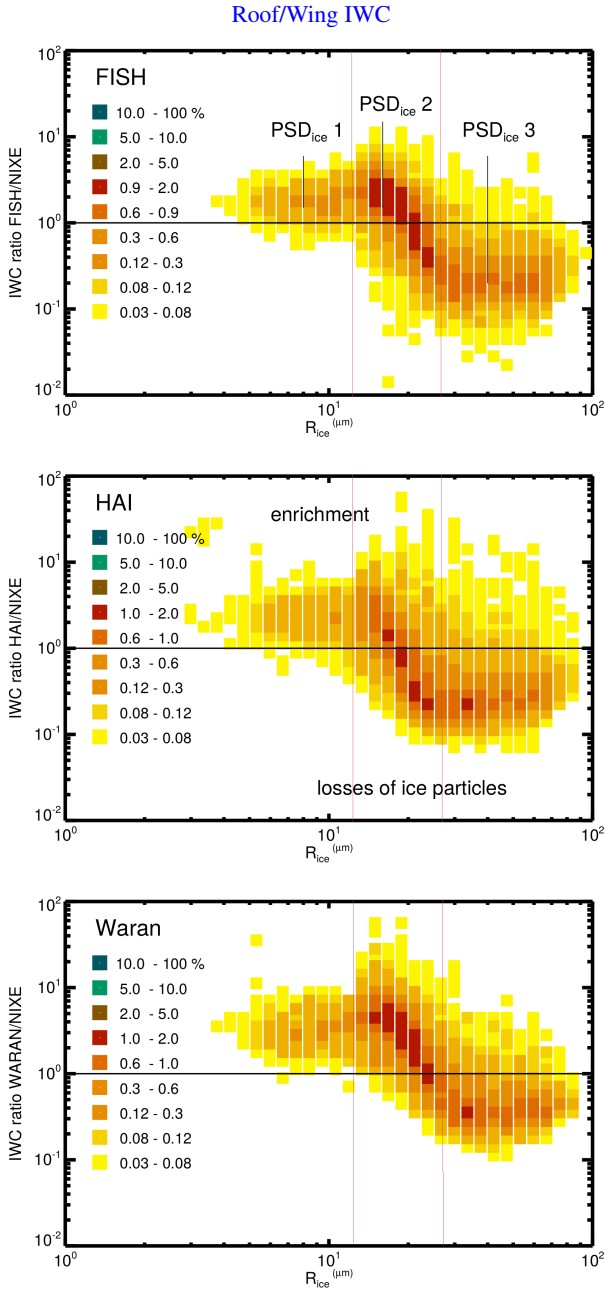

**Figure 12.** Ratios of Roof/Wing IWC (Roof IWC from FISH, HAI, Waran; Wing IWC from NIXE) vs. mean mass $R_{ice}$.

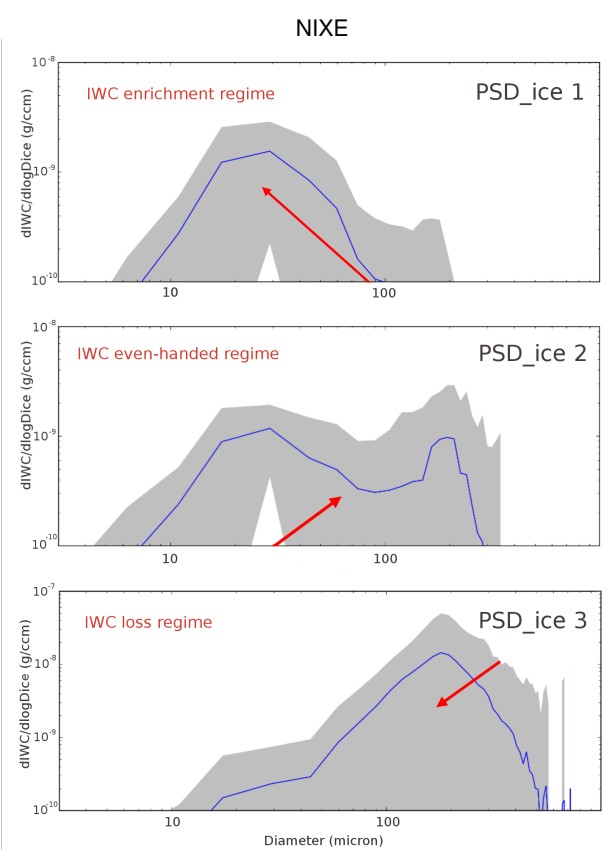

**Figure 13.** Three types of cirrus mass size distributions dIWC/dlogD$_{ice}$, exemplarily for the flight on 4. April 2014. Blue lines represent the mean PSDs, the grey area the standard deviation; note that for the portrayal of the PSD we use the ice particle diameter and not radius to clearly distinguish from the mean mass radius R$_{ice}$ of the ice particle population used in Figures 12.

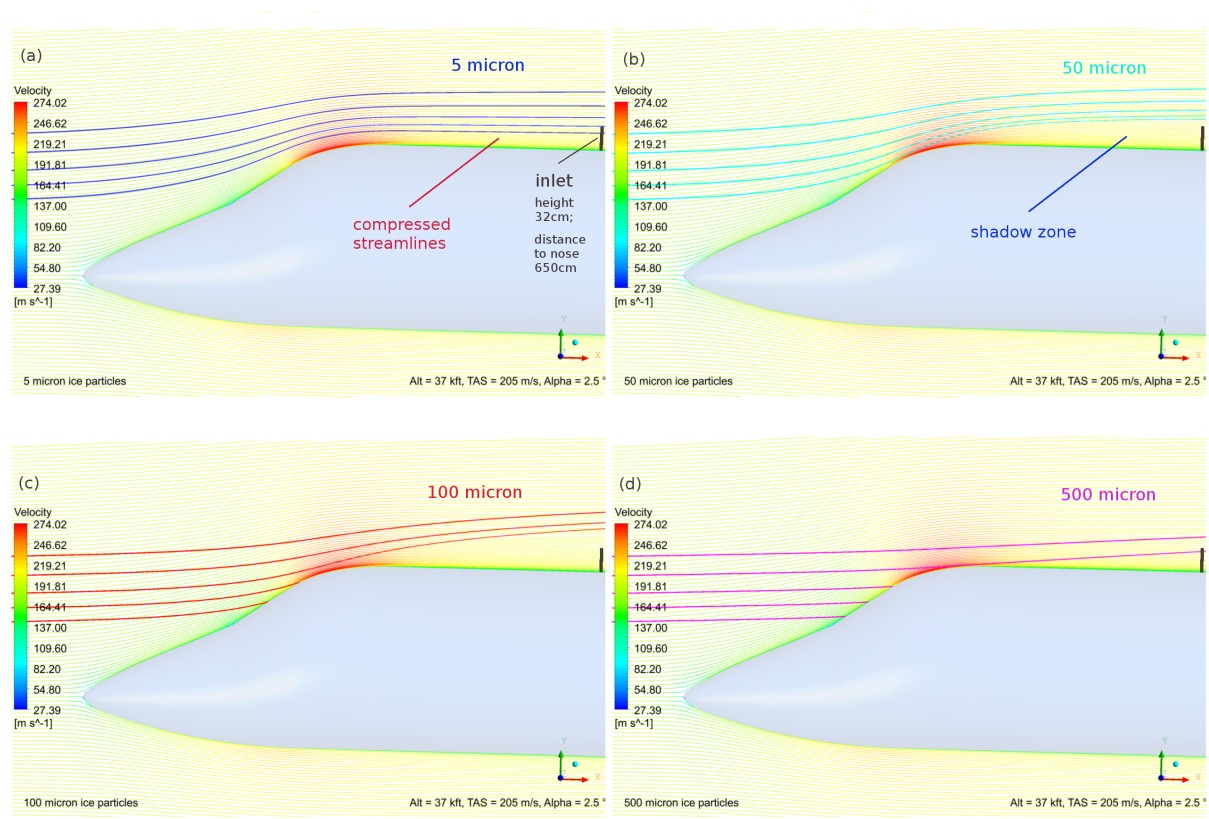

**Figure 14.** Two-dimensional CFD calculations of gas streamlines (thin lines, color coded by the velocity of the flow) and ice particle trajectories (thick colored lines) around an aircraft with a fuselage similar to the HALO aircraft (note that for legal reasons, the exact envelope of HALO can not be simulated). The IWC inlet is placed at the roof at the same position as the TGI on HALO (see Figure 4, Section 4.1.1). The simulations are performed for typical conditions during penetrations of cirrus clouds: Altitude = 37 kft, true air speed TAS = 205 m/s, angle of attack AOA = 2.5° and an ice crystal density of 0.918 g/cm$^3$. The panels are for different particle sizes, indicated in the panels. Ice particles starting at the lowest trajectory position enter the middle inlet tube of the IWC inlet if the particle follows the gas streamline. The simulations are performed by means of CFX 18.2 by ANSYS Inc., for a more detailed description of the methods applied in the simulations see Weigel et al. (2016).

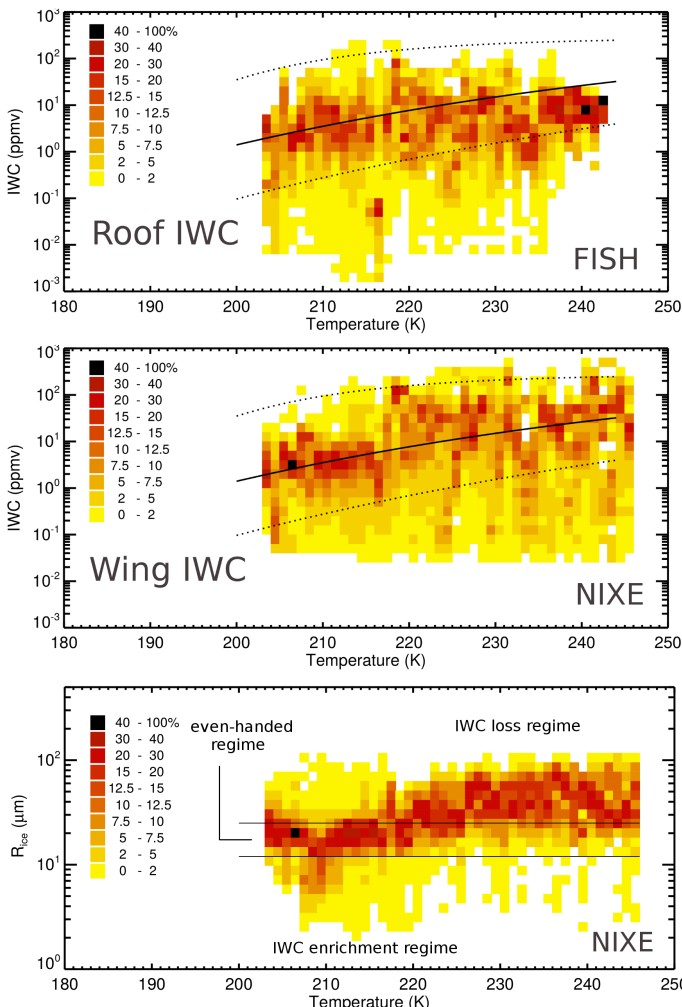

**Figure 15.** Top and middle panel: IWC and in dependence of temperature during ML-CIRRUS 2014, from roof-mounted FISH and wing-mounted NIXE (color code: frequencies of occurrence, black solid and dashed lines: median, core min. and max. IWCs after Schiller et al., 2008). Bottom panel: $R_{ice}$ in dependence of temperature during ML-CIRRUS 2014, from wing-mounted NIXE (the black lines denote the size regimes where ice particles are lost, enriched or both, for detail see Section 5.2).