# Peer review of "Ice particle sampling from aircraft – influence of the probing position on the ice water content"

_Atmospheric Measurement Techniques, 2017_

## Referee Comment (RC1) · Anonymous Referee #1 · 13 Nov 2017

Review of "Ice particle sampling from aircraft-Influence of the probing position on the ice water content." By Afchine et al.

Recommendation: Might be acceptable for publication pending mandatory major revision

The subject matter of this paper is very timely and appropriate for AMT, examining how the positioning of microphysics probes affects the measured ice water content. The authors compare measurements of bulk ice water contents (IWCs) made on the roof of the aircraft fuselage against those attached under a wing of the HALO aircraft, and also compare measurements mounted on the fuselage side and bottom of the Geophysica. Based on their comparison, the authors claim that the reason the IWCs measured by the roof inlets deviate from those under the aircraft wing is caused by the

shadow-zone behind the aircraft cockpit. Although the authors do provide one good piece of evidence to justify their claims (their Fig. 10 that shows how the ratios of the roof/wing IWC with the mean ice crystal size), overall I found that many of the claims made in the manuscript were not well justified by the data presented and the authors did not consider all the nuances associated with the probe positioning. Provided that the authors are able to do a more thorough job of discussing the limitations of their findings and provide better justification of their conclusions, I feel that this paper should be ultimately accepted by AMT.

The first major problem with the manuscript is that the authors overly simplify the discussion of the flow around an aircraft and the impact of the positioning of the probe. Although their Fig. 1 (adapted from King 1984) is a great illustration of the conceptual flow around an aircraft, it is important to note that the King (1984) calculations show that for the three different aircraft shapes they examined that the width of the shadow zone and the concentration enhancement factors could be described in terms of the scaled fuselage radius and a parameter similar to the Stokes number. I did not see anywhere in this paper where the authors discussed the expected location of the shadow or enhancement zones based on the fuselage radius or this "Stokes parameter" for either the HALO or Geophysica. It would seem that such a calculation would be required to justify their conclusions. Note, that ideally a complete flow analysis around the aircraft would be completed, but I am aware that such an analysis would be well beyond the scope of the paper. However, this later calculation would be something well within what would be expected for the scope of this paper.

Second, the authors make no comments about the position of the probe away from the fuselage, above the roof, or underneath the wind of the aircraft. Knowing this location is very critical to determining whether the probe is in an enhancement or shadow zone. For example, probes underneath an aircraft wing that are either not far enough below the wind or not far enough ahead of the leading edge of the wing might also suffer some large effects from the flow around the aircraft. The authors have included no

discussion about this whatsoever in their paper. First, I think more details about the mounting location of the probes are required, and this position should be assessed in terms of the expected location of the shadow/enhancement zones from the King (1984) type analysis.

Third, I see that the scatter between the IWC derived from the side and wing-mounted probes of a factor of 2.5 quite large. Further, the scatter can go even beyond this mean 2.5 figure. This seems extremely large to me. What is the uncertainty of the IWCs that are measured by these probes? Is it as large as a factor of 2.5? If less, what is causing the large amount of scatter? There needs to be more thorough uncertainty analysis than is currently presented in the paper.

Fourth, the authors attribute the differences they are seeing to the locations of the different probes. I agree that this seems to be the most likely reason for the differences. But, it would seem that to properly attribute this to the location of the probes, experiments should have been performed where the probes were switched between the different positions to see that the same order of differences still occurred. Is such a switch possible given the mounting possibilities on the aircraft?

Fifth, the authors need to do a better job in characterizing the uncertainty associated with the derivation of IWCs associated from size-resolved measurements through the use of m-D relations. Whereas the authors do acknowledge these uncertainties, noting that the bulk IWC is less error-prone in comparison to the IWC from the PSD, I feel that they are rather premature in making the claim that their m-D relation has demonstrated the robustness of their connection between cirrus ice crystal size and mass. This again seems a bit suspect given the difference of a factor of plus or minus 2.5. This also seems quite large compared to some other studies that have studied the variance of m-D relations in how they are related to calculations of bulk IWC and comparison with that derived from size distributions. How would the use of other m-D relations compare? Would some also work within the 2.5 factor or would they be smaller/larger? These issues need to be addressed especially if a conclusion is going to be made

that "the agreement of the IWCS ... demonstrates the validity of the m-D relation of Erfani and Mitchell (2016), slightly modified by Kramer et al. (2016) and Luebke et al. (2016)." There can be variations in m-D relations as the particle habits and densities can change not only with temperature, but also with the type of cloud being sampled.

In addition to the effect of the flow around the aircraft, there can also be impacts due to the geometry of the probe itself and the flow around the probe. For example, due to the flow around the CAPS probe, there are pressure perturbations around this probe that also might exist that could perhaps cause some flow distortions. Further, no comments were made about bouncing off the plane surface. Depending on where the probes are mounted underneath the wings, there can also be bouncing off of particles that can affect the measurements.

Finally, I think it is also very important that the conclusions made are specific. The authors may want to claim for that the particular probes mounted on the specific aircraft at the specific locations, there are certain things that can be said about preferred mounting locations. However, there simply is not sufficient evidence to generalize these findings to mounting locations on aircraft in general, or to locations in general (below wing, on roof, on fuselage, etc.)

Other Comments:

Page 5, line 19, What is sufficient distance?

Page 7, line 30: Was there any precipitation probe? What did the mass distribution function look like? Is there any possibility some mass is being missed in the IWC from the lack of particles above 937 micrometers being measured? Even if such particles are contributing minimally to the number, they can contribute more substantially to the mass.

---

## Referee Comment (RC2) · Anonymous Referee #2 · 26 Nov 2017

General Comments:

Measurements of ice water content at different aircraft mounting locations are potentially of interest, since much of current knowledge is based on potential flow or CFD models. It's a start, but this paper needs major revisions before it is publishable. The presentation is confusing, and much of the introductory material (including objectives) lacks focus and clarity. Also, the paper seems incomplete without additional work that is needed to quantify and scale each fuselage position for different aircraft. My specific suggestions are below.

Specific Comments:

1) There is very awkward English used throughout. Please avail yourself of an English

editing service.

2) Abstract line 3-8: Please clearly explain that you are comparing upper fuselage vs wing measurements on one business-jet aircraft from one experiment, and separate comparisons on specialized high-altitude aircraft from different experiments. The various aircraft wing and cockpit geometries are very different, and not everyone will know that HALO is a Gulfstream G-V, or what the Geophysica and WB57F are.

3) Abstract line 20: A "factor of 2.5" doesn't sound like good agreement, and may be misleading as actually the vast majority of your data points are much better than that. I recommend finding a better way of quantifying the data comparisons (see also point 17).

4) Page 2, line 1-2: Or "solid measurements" could also be made with an instrument mounted in a wingpod with extending inlet.

5) line 18: You can and should discuss the width of the shadow zone for each aircraft, based on the King (1984) modified Stokes parameter. Granted this is an estimate, but it will give an idea of the expected variance for different aircraft fuselage sizes and stations (distance back) on the aircraft. Ice crystal sizes and can be converted to aerodynamic diameters and modified Stokes parameter for typical crystal sizes and shapes.

6) lines 20-25: As in the Abstract, what measurements are being compared on which aircraft is confusing. You can't necessarily generalize from one aircraft to another. Please be specific.

7) Page 3, line 24: Every inlet will influence the airflow somewhat. So, switch "not influence" to "minimally influence".

8) Page 4, Section 2.1.2: Not sure that all this detail is required; you could just specify the uncertainty/detection limits for each instrument and reference papers for more information.

9) lines 18-23: If you are going into all this detail, a figure would be helpful. Or the section could be cut.

10) Lines 31-32: Only if the flow rate is not controlled, which it can be in some flow configurations.

11) Page 5, line 1: This seems backwards, since you are solving for IWC.

12) Line 8; too much detail; not sure why all this is worthy of note for this paper.

13) Line 21, insert "for particle measurements" after "flow around wings", as obviously the airflow is critical for other things (like lift).

14) Page 6: line 13-14: A philosophical point: it's already known that the top of the fuselage is a bad place to sample clouds, so why were all these instruments mounted here? Are they primarily to measure gas-phase composition, with cloud measurements just for this study?

15) Specify the distances from fuselage and fuselage station (distance back) for each inlet position.

16) Page 8: line 13-14: but HAI is actually closer to fuselage, right? How much?

17) Line 15: Actually it seems only a small fraction of measurements differ by 2.5. This should be reworded for quantitatively, and to make it clear that actually most data that fall within smaller ratios. And many of these are at small IWCs, and likely influenced by higher uncertainties at low values (due to subtracting a relatively large clear air signal, and possibly calibration uncertainties). This should be discussed. Likewise with the factor of 10 later on. Also, are the data from different instruments synced precisely?...as this can also increase scatter.

18) It would also be nice to know if the different instruments have been successfully compared in the lab, a wind tunnel or in past aircraft campaigns.

19) Page 9: Again, we need to know how far out and back each inlet is.

20) Lines 10-11: The Geophysica is also a narrower aircraft. Cannot compare directly with the G-V without scaling somehow.

21) Lines 15-16: Page 10: lines 8 on: This is interesting, but it should be clarified that at very large sizes, particle trajectories are straight and little enhancement or shadowing is expected (ie, high S values for King, 1984). It appears this is outside the range of what you sampled, although it's difficult to know since S values aren't calculated.

22) Page 12: Lines 7-8: This is simplistic and dangerously misleading, since there is still a shadow zone on the side and bottom of the fuselage–it's just more narrow than on the top. It also will vary with fuselage diameter and distance behind the nose. Again, precise inlet locations are needed.

23) Need to reference prior work. Lines 15-16: Twohy and Rogers (J. Atmos. Ocean. Tech, 1993) also reported deviations in measured cloud properties for different aircraft mounting locations. Lines 18-20: Davis et al (JGR, 2007) also compared IWC measurements on the WB57F.

---

## Referee Comment (RC3) · Anonymous Referee #3 · 6 Dec 2017

Evaluation of the paper titled: "Ice particle sampling from aircraft – influence of the probing position on the ice water content" by Afchine et al.

Overview: This work examines the important problem of the effects of a probe's location on an aircraft, on the accuracy of its measurements. This topic has a long history of research, and has been explored since the mid-70s by different groups (e.g. Norment and Zalosh, 1974). The most well-known studies in this area were published in a series of papers by Warren King in the mid-80s. Based on the theoretical analysis of particle trajectories followed by in-situ verification (King, 1984; King et al. 1984), it was concluded that the particle number and mass concentrations can be biased by an order of hundreds of percent depending on the mounting location of the probe on the fuselage of the airplane. One of the important outcomes of the King's studies is the identification of the regions with enhanced and reduced concentrations of cloud particles at the top of the fuselage. The most favorable places for bulk microphysical instrumentation installation on the fuselage would be the side and bottom positions. This rule has been followed by many research groups when instrumenting research aircrafts for cloud microphysical measurements. The present study reiterates King's conclusion, that the cloud microphysical measurements (specifically IWC) at the side and bottom fuselage locations are more accurate compared to the top location. So, in this regard, this study confirms the existing knowledge about the preferential fuselage locations of the bulk microphysical instruments. In the present work, the conclusion about the accuracy of IWC measurements was obtained based on the comparisons of the TWC probes mounted on the different fuselage locations: top, side and bottom. Even though I agree with the conclusions of this paper, the methodology of the approach employed in this study leaves many questions unanswered. Additionally, critical components of the study of the probing positions are missing: for example, there is no assessment of the dimension of the shadow zone and its distance from the fuselage, the effect of the air density of the particle trajectories and size of the shadow zone is not accounted for, the ice concentration enhancement around the fuselage due to ice bouncing is not accounted for, the particle trajectory analysis has been omitted.

In my opinion, this study should be eventually published. However, in its present form the paper is not suitable for publication in AMT. At this stage I would suggest withdrawing the manuscript and adding the missing necessary components. Because of the great importance of the considered question, and the large anticipated impact of this work on the cloud instrumentation community, I would encourage the authors to address the questions listed below and resubmit the manuscript.

Major comments:

1. This paper validates the conclusion of the King et al (1984) study on a different instrumental basis. Further progress can be achieved by utilizing flow simulations and particle trajectory analysis. At present, CFD analysis is routinely used by different research groups (especially in the aviation community) to analyze the particle trajectories for different aspects of aviation safety and to study the accuracy of measurements of cloud microphysical instrumentation (e.g. Weigel et al., AMT, 2017). It would be highly beneficial for this paper to include these types of simulations. This will help in addressing many aspects of the positioning of the cloud microphysical instrumentation, and provide estimates of the accuracy of measurements. The CFD and particle trajectory analysis may take some time. However, the obtained results will be rewarding for the community.

2. The dimensions of the shadow and enhancement zones at the mounting location of the TWC probes of the HALO aircraft should be provided here. At that stage it is not clear whether the TWC inlets were located inside the shadow zone, enhancement zone or in the relatively undisturbed free flow. Without such information, the discussion is incomplete.

3. King (1984, part 1) considered the formation of the shadow zone on the top of the fuselage for liquid droplets. Liquid droplets after the impact with the fuselage stick to its surface and shed downstream (see Fig.6 in King, 1984, part 1). However, ice particles after impact with the fuselage rebound back into the airflow. Ice particles, after the first rebound, may experience multiple bouncing. This phenomenon was observed in wind tunnels and is well reproduced in CFD simulations (e.g. Korolev et al JTECH, 2013). One of the consequences of this effect is an enhanced concentration of ice particles around the fuselage including side and bottom locations. This is results in a principal difference compared to the King's (part 1 and 2) work, which was focused on the trajectories of liquid droplets. In this regard, it is important to consider the enhancement of ice concentration not only at the top of the fuselage, but all the way around it. This effect may equally affect IWC measurements at the side and bottom locations. This question should be properly addressed.

4. CFD simulations showed that particle trajectories are sensitive to air density air. Therefore, the dimensions of the shadow and enhancement ice particle zones depend

on the air density air along with other parameters such as TAS, AoA, etc. This is a very important issue and it should be properly addressed in this study. Could you also comment on the effect of air on the dependences of IWC ratio vs Rice shown in Fig.10?

5. Page 10. The equation mean mass radius Rice = IWC/Nice should be written as Rice = (3IWC/4$\pi$ice Nice)^1/3. I believe this a typo. Unfortunately, no information about ice was provided in the text. Since the size-to-mass parameterization M=aRice^b was applied for the IWC calculation, then ice is a function of Rice, i.e. ice = 3aRice^(b-3)/4$\pi$. Therefore, the mean mass radius should be calculated as Rice = (IWC/aNice)^1/b. Could you please clarify how Rice was calculated?

6. It is important to indicate the distance of the TWC probes inlets from the fuselage and from the nose of the airplane. This is necessary to understanding the effect of the probe's location on the accuracy of its measurements. Along this way, it would be beneficial to include a summary table with the positioning of the TWC probes, type of the airplane, name of the project, TWC probe, particle probe used as a reference, etc.

7. The diagrams in Figs 7, 8, 9 in their present form visualize the scattering of the IWC points. However, it is difficult to judge about the biases and the degree of scattering of the data points. It is suggested to add a linear regression line, indicate a relevant linear equation, standard deviation, and correlation coefficient in each diagram. This information will help to quantify of the degree of agreement between the IWC measurements. Please also provide the averaging time used for the data these diagrams.

8. The IWC calculated from the cloud particle probes (CAS-DPOL, CIP-G,2DS) was used as a reference for the TWC probes (FISH, HAI, Waran) measurements. The processing of the scattering and imaging probes are sensitive to the algorithms and assumptions employed in the processing software. Thus, CAS-DPOL is usually calibrated in assumption that the cloud particles are spherical water droplets. Were any corrections for ice applied for the CAS-DPOL data? What algorithms and corrections were used during the processing of the 2D probe's data? What are the typical, min, and

max number of particles in the CAS, CIP and 2DS data? Please provide an assessment of the statistical significance of PSDs used for the IWC calculations. Statistically insignificant PSDs may result in large random errors in IWC calculations. These questions should be elaborated upon and explained in the text. The assessment of the errors in the IWC calculations for the particle probe data should be provided as well.

9. The diagrams in Fig.10 are supportive of the statement about oversampling of small particles and undersampling of large particles at the roof location. Similar diagrams should be provided for the side and bottom locations of the TWC inlets on Geophysica and WB57. Otherwise, one could argue that the 'duck' type behavior of the IWC ration vs Rice is a result of the errors in calculations of IWC from the particle probes.

10. Page 4, Line 15: "However, isokinetic sampling (= the flow inside the inlet is the same as in the free flow), which in principle enables the undisturbed measurement of H2Otot, is not possible for fast flying aircraft, since the air flow speed is always much higher than the velocity inside of the inlet." The airborne version of the isokinetic probe for measurements of cloud condensed water was designed by NRC: (Davison, C., J. MacLeod, J. Strapp, and D. Buttsworth, 2008: Isokinetic total water content probe in a naturally aspirating configuration: Initial aerodynamic design and testing. Proc. 46th AIAA Aerospace Sciences Meeting and Exhibit, Reno, NV, American Institute of Aeronautics and Astronautics, AIAA 2008-435. [Available online at http://arc.aiaa.org/doi/abs/10.2514/6.2008-435.]) This probe was successfully operated during several field campaigns on different aircrafts. Some results were published in JTECH.

11. Traditionally, condensed water content is measured in g/m3 (liquid, ice or total water content) or g/kg (mixing ratio). These units are well adapted by the cloud and climate modeling communities (both g/m3 and g/kg), remote sensing community (g/m3), aviation industry (mainly g/m3). The present paper is utilizing non-conventional units in the cloud physics community (ppmv) in order to describe condensed water content. This unit is usually used to describe concentration of a gas phase, rather than to characterize the weight of a liquid or solid phase per unit volume. This unit is mainly employed by the subcommunity formed around the evaporators used for measurements of the condensed cloud phase (e.g. FISH, HAI, Waran, etc.). I am not sure that employing this unit adds clarity; rather, it creates barriers in the dissemination of the IWC measurements that employ this unit. In my opinion, the cloud and climate modeling communities and the remote sensing community are unlikely to switch to this unit. The aviation industry is quite conservative, and it most likely they will ignore the measurements of condensed water content in this unit. I recommend using the conventional units of g/kg or g/m3. At minimum, I suggest using additional axes with conventional units in Fig. 7, 8 ,9, 10.

Minor comments:

1. Page 2, Line 11: "The IWC of a cirrus is a bulk quantity which is composed of the sum of all ice particles. . ." The term "of a cirrus" is redundant here. This statement is relevant to any cloud, not just cirrus.

2. Page 2, Line 11: It should be ". . .the sum of all ice particles masses. . ."

3. Page 2, Line 15: "In particular, King (1984) shows that above the roof of an aircraft the sampling of particles is disturbed." Strictly speaking, the sampling of particles is disturbed everywhere around the fuselage. However, the scale of this disturbance is different. Please reword this sentence.

4. Page 2, Line 16: "However, to simulate and quantify losses or enrichment of ice particles and the effect on IWC at a specific position of an aircraft is hardly possible, since this depends on the prevailing particle size distribution and also the irregular shape of the ice crystals." This is a too strong of a statement. The irregular ice particle shapes can be replaced with spheres with equivalent aerodynamic size. For example, particle trajectory analysis can be performed using spheres with the mass density calculated from size-to-mass parameterization M=aD^b.

5. Page 2, Line 27: "The IWC of cirrus can be recorded from aircraft either by bulk cloud measurements using airborne closed path hygrometers mounted behind an inlet tube or via integration of the ice particle number size distributions (PSDice) measured by cloud spectrometers. In both cases, the ice particles must be properly sampled before the measurement." Hot-wire probes are missed in this statement.

6. Page 2, Line 29: "The bulk IWC is less error prone in comparison to the IWC from PSDice in case of an undisturbed measurement." This is a questionable statement. Both techniques have its own problems and advantages.

7. Page 3, Line 1: replace "Fore" to "For".

8. Page 3, Line 18: "To precisely detect H2Otot" replace by "To precisely measure H2Otot"

9. Page 4, Line 6: "To specify the size ranges of the 'smaller' and 'larger' cloud particles, CFD calculations for the specific conditions of fuselage shape, aircraft speed and inlet distance from the nose of the aircraft need to be performed." This sentence is disconnected from the following text and it appears to be redundant.

10. Page 4, Line 7: "Very roughly, cloud particles with radii <30 $\mu$m can be assumed to belong to the smaller, while those >30 $\mu$m are associated to the larger part of the cloud particle size spectrum at jet aircraft with high air speeds." What is the basis for this statement? References should be provided here.

11. Page 4, Line 23: "...shattering into small artifacts at the cloud probes head..." should be "...shattering into small fragments at the cloud probes' housing..."

12. Page 4, Line 23: "However, for the calculation of the IWC, the uncertainty from shattering does not play a significant role since the shattered crystals still contribute to the integrated mass of PSDice." This sentence should be reworded.

13. Page 4, Line 9: IWCS should be IWCs

14. Figure 11. The y-labels are not easily legible. Please enlarge the font size.

---

## Author Comment (AC1) · 17 Mar 2018

Answer to the referees comments on

**Ice particle sampling from aircraft –**
**influence of the probing position on the ice water content**

by

Afchine, A., Rolf, C., Costa, A., Spelten, N., Riese, M., Buchholz, B., Ebert, V., Heller, R., Kaufmann, S., Minikin, A., Voigt, C., Zöger, M., Smith, J., Lawson, P., Lykov, A., Khaykin, S., and Krämer, M.

*Correspondence to:* Martina Krämer (m.kraemer@fz-juelich.de)

First of all, a great thank to all three referees for their thorough reading of the paper and their detailed comments, which helped to improve the manuscript.

We have taken into account all the points noted by the referees and included almost all of them in the new manuscript.

Reading the reports we understood that the description of the method we used and also the intention of the study was to brief and thus not clear. Hence, many of the changes in the manuscript are adding more detailed information. In particular, we have added a new section (Section 2: Methodoly) describing the approach of the study.

Further, to confirm the results from the approach of comparative IWC measurements, we have added CFD simulations around the HALO aircraft for different ice particle sizes, as desired by all referees (new Figure 14).

Our point by point answers to referees are in blue.

Answer to Ref. #1:

The subject matter of this paper is very timely and appropriate for AMT, examining how the positioning of microphysics probes affects the measured ice water content. The authors compare measurements of bulk ice water contents (IWCs) made on the roof of the aircraft fuselage against those attached under a wing of the HALO aircraft, and also compare measurements mounted on the fuselage side and bottom of the Geophysica. Based on their comparison, the authors claim that the reason the IWCs measured by the roof inlets deviate from those under the aircraft wing is caused by the shadow-zone behind the aircraft cockpit. Although the authors do provide one good piece of evidence to justify their claims (their Fig. 10 that shows how the ratios of the roof/wing IWC with the mean ice crystal size), overall I found that many of the claims made in the manuscript were not well justified by the data presented and the authors did not consider all the nuances associated with the probe positioning. Provided that the authors are able to do a more thorough job of discussing the limitations of their findings and provide better justification of their conclusions, I feel that this paper should be ultimately accepted by AMT.

1. The first major problem with the manuscript is that the authors overly simplify the discussion of the flow around an aircraft and the impact of the positioning of the probe. Although their Fig. 1 (adapted from King 1984) is a great illustration of the conceptual flow around an aircraft, it is important to note that the King (1984) calculations show that for the three different aircraft shapes they examined that the width of the shadow zone and the concentration enhancement factors could be described in terms of the scaled fuselage radius and a parameter similar to the Stokes number. I did not see anywhere in this paper where the authors discussed the expected location of the shadow or enhancement zones based on the fuselage radius or this 'Stokes parameter' for either the HALO or Geophysica. It would seem that such a calculation would be required to justify their conclusions. Note, that ideally a complete flow analysis around the aircraft would be completed, but I am aware that such an analysis would be well beyond the scope of the paper. However, this later calculation would be something well within what would be expected for the scope of this paper.

   We have performed now CFD calculations (showing the enrichment and shadow zones of the HALO aircraft) that justifies our conclusions – see new Figure 14 and extended Section 5.2 (old section 4.2).

2. Second, the authors make no comments about the position of the probe away from the fuselage, above the roof, or underneath the wind of the aircraft. Knowing this location is very critical to determining whether the probe is in an enhancement or shadow zone. For example, probes underneath an aircraft wing that are either not far enough below the wind or not far enough ahead of the leading edge of the wing might also suffer some large effects from the flow around the aircraft. The authors have included no discussion about this whatsoever in their paper. First, I think more details about the mounting location of the probes are required, and this position should be assessed in terms of the expected location of the shadow/enhancement zones from the King (1984) type analysis.

We have included the exact positions of the probes into Section 4 (old Section 3). A discussion on effects that can influence IWC measurement (except of roof sampling) is introduced in the new Sections 2 (Methodology).

3. Third, I see that the scatter between the IWC derived from the side and wing-mounted probes of a factor of 2.5 quite large. Further, the scatter can go even beyond this mean 2.5 figure. This seems extremely large to me. What is the uncertainty of the IWCs that are measured by these probes? Is it as large as a factor of 2.5? If less, what is causing the large amount of scatter? There needs to be more thorough uncertainty analysis than is currently presented in the paper.

A more detailed discussion is now presented in new Section 5.1.4 (Scatter of IWC measurements).

4. Fourth, the authors attribute the differences they are seeing to the locations of the different probes. I agree that this seems to be the most likely reason for the differences. But, it would seem that to properly attribute this to the location of the probes, experiments should have been performed where the probes were switched between the different positions to see that the same order of differences still occurred. Is such a switch possible given the mounting possibilities on the aircraft?

The referee is right that switching instruments at the same plane would be the ultimate proof of the conclusion that the roof sampling position is causing the differences in the IWC measurement. Unfortunately, this is impossible on HALO. However, on board of Geophysica, we operated the same instrumentation (FISH and NIXE-CAPS) as on HALO, but with side mounting for the total water measurement (this is now mentioned in the Section 5.1.3 - old 4.1.3). Further, FISH is used on the WB-57 at the bottom of the plane, so we have the same instrument for the total water measurements on three aircraft (and two very similar cloud probes for the PSD$_{ice}$ measurements).

To better explain the method that we have used to evaluate and quantify differences our IWC measurements on aircraft, we have introduced a new Section (Section 2: Methodology).

5. Fifth, the authors need to do a better job in characterizing the uncertainty associated with the derivation of IWCs associated from size-resolved measurements through the use of m-D relations. Whereas the authors do acknowledge these uncertainties, noting that the bulk IWC is less error-prone in comparison to the IWC from the PSD, I feel that they are rather premature in making the claim that their m-D relation has demonstrated the robustness of their connection between cirrus ice crystal size and mass. This again seems a bit suspect given the difference of a factor of plus or minus 2.5. This also seems quite large compared to some other studies that have studied the variance of m-D relations in how they are related to calculations of bulk IWC and comparison with that derived from size distributions. How would the use of other m-D relations compare? Would some also work within the 2.5 factor or would they be smaller/larger? These issues need to be addressed especially if a conclusion is going to be made that 'the agreement of the IWCS . . . demonstrates the validity of the m-D relation of Erfani and Mitchell (2016), slightly modified by Kramer et al. (2016) and Luebke et al. (2016).' There can be variations in

m-D relations as the particle habits and densities can change not only with temperature, but also with the type of cloud being sampled.

We have added a Figure (new Figure 8) to new Section 4.2 (old 3.2: Cloud spectrometers for IWC) showing that the use of other m-D relations will not greatly alter the IWC calculations. We think that Figure 8 together with the discussion of the scatter and significance of IWC measurements in the new Sections 2 and 5.1.4 now makes the conclusions drawn in the paper more sound.

6. Finally, I think it is also very important that the conclusions made are specific. The authors may want to claim for that the particular probes mounted on the specific aircraft at the specific locations, there are certain things that can be said about preferred mounting locations. However, there simply is not sufficient evidence to generalize these findings to mounting locations on aircraft in general, or to locations in general (below wing, on roof, on fuselage, etc.)

We have deleted generalizing statements in the manuscript. However, I do not fully agree with the referee here. What we have shown with the measurements -and now also with CFD calculations- confirm the current knowledge from experimental and theoretical work, that means it is not only very specific for these aircraft or instrumentation. What is specific here are the size ranges and strength of ice particle enrichment or losses on the planes roof.

Other Comments:

7. Page 5, line 19, What is sufficient distance?

' ... sufficient distance ... to minimize particle losses or enrichment due to distorted cloud particle trajectories ...'
The distances for the deployed instrument are now given in Table 1.

8. Page 7, line 30: Was there any precipitation probe? What did the mass distribution function look like? Is there any possibility some mass is being missed in the IWC from the lack of particles above 937 micrometers being measured? Even if such particles are contributing minimally to the number, they can contribute more substantially to the mass.

No, the was no precipitation probe on board. And as can be seen from Figure 13, in most cases the particle sizes do not exceed 1000 $\mu$m - which is typical for cirrus clouds. In cases with larger ice crystals you are right - then the losses in IWC from $PSD_{ice}$ are even larger. These cases happen mostly at the warmer cirrus temperatures ($> 230$ K), where liquid origin cirrus dominates.

Answer to Ref. #2:

General Comments:

Measurements of ice water content at different aircraft mounting locations are potentially of interest, since much of current knowledge is based on potential flow or CFD models. It's a start, but this paper needs major revisions before it is publishable. The presentation is confusing, and much of the introductory material (including objectives) lacks focus and clarity. Also, the paper seems incomplete without additional work that is needed to quantify and scale each fuselage position for different aircraft. My specific suggestions are below.

Specific Comments:

1) There is very awkward English used throughout. Please avail yourself of an English AMTD editing service.

We will do this at the final stage of the production process by AMTD.

2) Abstract line 3-8: Please clearly explain that you are comparing upper fuselage vs wing measurements on one business-jet aircraft from one experiment, and separate comparisons on specialized high-altitude aircraft from different experiments. The various aircraft wing and cockpit geometries are very different, and not everyone will know that HALO is a Gulfstream G-V, or what the Geophysica and WB57F are.

We have specified the aircraft types of the different field experiments.

3) Abstract line 20: A 'factor of 2.5' doesn't sound like good agreement, and may be misleading as actually the vast majority of your data points are much better than that. I recommend finding a better way of quantifying the data comparisons (see also point 17).

The factor of 2.5 is explained in more detail in the new manuscript in Section 4.1.4.

4) Page 2, line 1-2: Or 'solid measurements' could also be made with an instrument mounted in a wingpod with extending inlet.

This sentence is deleted.

5) line 18: You can and should discuss the width of the shadow zone for each aircraft, based on the King (1984) modified Stokes parameter. Granted this is an estimate, but it will give an idea of the expected variance for different aircraft fuselage sizes and stations (distance back) on the aircraft. Ice crystal sizes and can be converted to aerodynamic diameters and modified Stokes parameter for typical crystal sizes and shapes.

We have introduced a new section (new 2. Methodology) where we discuss in more detail the approach we used here to address the quality of IWC measurements at different probing positions. In this section and, more detailed in Section 5.2 (old section 4.2), it is shown that from deviations of IWC measurements at the fuselage in comparison to wing IWC measurements it is possible to draw conclusions on the manner of possible IWC distortions, for example if the probing position is placed in a shadow or enrichment zone.

Since the method of comparative IWC measurements shows whether an inlet is placed in the shadow zone, without determining specifically the width of the zone of the aircraft, we feel that it is beyond the scope of this work to discuss the width of the shadow zone for each aircraft geometry.

Nevertheless, we have calculated the modified Stokes parameter provided by King (1984). Unfortunately, we found that the calculations are not applicable for the high cruising speeds of the planes involved in this study, since the angle of attack is not considered in that formulation.

Instead, we have included exemplary CFD simulations of gas streamlines and ice particle trajectories of different sizes around the HALO aircraft (new Figure 14) to demonstrate that our findings about shadow and enrichment zones using comparative IWC measurements are confirmed by the simulations.

6) lines 20-25: As in the Abstract, what measurements are being compared on which aircraft is confusing. You can't necessarily generalize from one aircraft to another. Please be specific.

See answer to 5)

7) Page 3, line 24: Every inlet will influence the airflow somewhat. So, switch 'not influence' to 'minimally influence'.

Done.

8) Page 4, Section 2.1.2: Not sure that all this detail is required; you could just specify the uncertainty/detection limits for each instrument and reference papers for more information.

It might be that this detail is not entirely essential. On the other hand, to my knowledge the relation between the enhanced total water and IWC and also detection limits are not described in detail elsewhere, and since we aim to do that we prefer to keep this section in the manuscript.

9) lines 18-23: If you are going into all this detail, a figure would be helpful. Or the section could be cut.

Thank you for this suggestion, we have added new Figure 2.

10) Lines 31-32: Only if the flow rate is not controlled, which it can be in some flow configurations.

We have noted that now.

11) Page 5, line 1: This seems backwards, since you are solving for IWC.

We have removed the confusing part of the sentence.

12) Line 8; too much detail; not sure why all this is worthy of note for this paper.

We added the following sentence to better explain why this detail is necessary: 'Because of this, the IWC detection limit as well as the uncertainty of IWC improves with decreasing temperature.'

13) Line 21, insert 'for particle measurements' after 'flow around wings', as obviously the air-flow is critical for other things (like lift).

Done.

14) Page 6: line 13-14: A philosophical point: it's already known that the top of the fuselage is a bad place to sample clouds, so why were all these instruments mounted here? Are they primarily to measure gas-phase composition, with cloud measurements just for this study?

The inlets are primarily mounted at the top for gas-phase measurements. Unfortunately, when the aircraft was first deployed, no special cloud inlets were installed, so here we would like to quantify the impact of the probing position on the IWC.

15) Specify the distances from fuselage and fuselage station (distance back) for each inlet position.

We have inserted Table 1 listing all distances.

16) Page 8: line 13-14: but HAI is actually closer to fuselage, right? How much?

Yes HAI is 5.5cm closer to the fuselage - see Table 1.

17) Line 15: Actually it seems only a small fraction of measurements differ by 2.5. This should be reworded for quantitatively, and to make it clear that actually most data that fall within smaller ratios.

We have drawn the 2.5 lines in new Figure 9 so that it can be better seen now that most of data points are within that range.

And many of these are at small IWCs, and likely influenced by higher uncertainties at low values (due to subtracting a relatively large clear air signal, and possibly calibration uncertainties). This should be discussed. Likewise with the factor of 10 later on.

The uncertainties are smaller at lower IWC, because the clear air signal decreases with temperature, see new Figure 3 and Section 3.1.2.

Also, are the data from different instruments synced precisely?...as this can also increase scatter.

The different instruments are precisely synced.

18) It would also be nice to know if the different instruments have been successfully compared in the lab, a wind tunnel or in past aircraft campaigns.

Unfortunately we had not the chance to compare all instruments before. However, FISH has been compared to a number of other instruments at the cloud chamber AIDA (see Fahey et al., 2014) and on aircraft (see Rollins et al., 2014). A review of two decades of successful measurements with FISH is given by Meyer et al. (2015). This is now mentioned in the manuscript at the beginning of new Section 5.1.1.

19) Page 9: Again, we need to know how far out and back each inlet is.

This is now given in new Table 1, which is also mentioned in the text.

20) Lines 10-11: The Geophysica is also a narrower aircraft. Cannot compare directly with the G-V without scaling somehow.

In the new Section 2 (Methodology) we have now better described the approach of comparative IWC measurements. One advantage of this approach is that all effects that influence IWCs, the aircraft shape and inlet positions included, are contained in the measurements.
Hence it seems very unlikely that the different widths of the aircraft is the reason of the derivations of the IWCs, especially when at the same time these derivations look as expected if an inlet is placed at the aircraft's roof.

21) Page 10: lines 8 on: This is interesting, but it should be clarified that at very large sizes, particle trajectories are straight and little enhancement or shadowing is expected (ie, high S values for King, 1984). It appears this is outside the range of what you sampled, although it's difficult to know since S values aren't calculated.

The CFD calculations we have performed include particles of 500 $\mu$m, see new Figure 14. The discussion in Section 5.2 is extended and includes these large ice crystals.

22) Page 12: Lines 7-8: This is simplistic and dangerously misleading, since there is still a shadow zone on the side and bottom of the fuselage – it's just more narrow than on the top. It also will vary with fuselage diameter and distance behind the nose. Again, precise inlet locations are needed.

That you for mentioning that - we have extended the recommendations to that effect.

23) Need to reference prior work. Lines 15-16: Twohy and Rogers (J. Atmos. Ocean. Tech, 1993) also reported deviations in measured cloud properties for different aircraft mounting locations. Lines 18-20: Davis et al (JGR, 2007) also compared IWC measurements on the WB57F.

Thanks again, these papers are referenced now in the manuscript (Davis et al., 2007, in the new Section 5.1.4 'Scatter of IWC measurements'; Twohey et al., 1993, in Section 3.1.1 'IWC enrichment or loss due to inlet position').

Answer to Ref. #3:

Overview: This work examines the important problem of the effects of a probe's location on an aircraft, on the accuracy of its measurements. This topic has a long history of research, and has been explored since the mid-70s by different groups (e.g. Norment and Zalosh, 1974). The most well-known studies in this area were published in a series of papers by Warren King in the mid-80s. Based on the theoretical analysis of particle trajectories followed by in-situ verification (King, 1984; King et al. 1984), it was concluded that the particle number and mass concentrations can be biased by an order of hundreds of percent depending on the mounting location of the probe on the fuselage of the airplane. One of the important outcomes of the King's studies is the identification of the regions with enhanced and reduced concentrations of cloud particles at the top of the fuselage. The most favorable places for bulk microphysical instrumentation installation on the fuselage would be the side and bottom positions. This rule has been followed by many research groups when instrumenting research aircrafts for cloud microphysical measurements. The present study reiterates King's conclusion, that the cloud microphysical measurements (specifically IWC) at the side and bottom fuselage locations are more accurate compared to the top location. So, in this regard, this study confirms the existing knowledge about the preferential fuselage locations of the bulk microphysical instruments. In the present work, the conclusion about the accuracy of IWC measurements was obtained based on the comparisons of the TWC probes mounted on the different fuselage locations: top, side and bottom. Even though I agree with the conclusions of this paper, the methodology of the approach employed in this study leaves many questions unanswered. Additionally, critical components of the study of the probing positions are missing: for example, there is no assessment of the dimension of the shadow zone and its distance from the fuselage, the effect of the air density of the particle trajectories and size of the shadow zone is not accounted for, the ice concentration enhancement around the fuselage due to ice bouncing is not accounted for, the particle trajectory analysis has been omitted.

In my opinion, this study should be eventually published. However, in its present form the paper is not suitable for publication in AMT. At this stage I would suggest withdrawing the manuscript and adding the missing necessary components. Because of the great importance of the considered question, and the large anticipated impact of this work on the cloud instrumentation community, I would encourage the authors to address the questions listed below and resubmit the manuscript.

Major comments:

1. This paper validates the conclusion of the King et al (1984) study on a different instrumental basis. Further progress can be achieved by utilizing flow simulations and particle trajectory analysis. At present, CFD analysis is routinely used by different research groups (especially in the aviation community) to analyze the particle trajectories for different aspects of aviation safety and to study the accuracy of measurements of cloud microphysical instrumentation (e.g. Weigel et al., AMT, 2017). It would be highly beneficial for this paper to include these types of simulations. This will help in addressing many aspects of the positioning of the cloud microphysical instrumentation, and provide estimates of the accuracy of measurements. The CFD

and particle trajectory analysis may take some time. However, the obtained results will be rewarding for the community.

We have included CFD simulations of gas streamlines and ice particle trajectories of different sizes around the HALO aircraft (new Figure 14, Section 5.2). Further, we have added a new section to the manuscript (Section 2: Methodology) to better explain the approach of comparative IWC measurements we used here. In this section, also the application of CFD simulations to evaluate especially IWC measurements are discussed.

2. The dimensions of the shadow and enhancement zones at the mounting location of the TWC probes of the HALO aircraft should be provided here. At that stage it is not clear whether the TWC inlets were located inside the shadow zone, enhancement zone or in the relatively undisturbed free flow. Without such information, the discussion is incomplete.

The shadow and enrichment zone of the HALO aircraft is shown in the new Figure 14.

3. King (1984, part 1) considered the formation of the shadow zone on the top of the fuselage for liquid droplets. Liquid droplets after the impact with the fuselage stick to its surface and shed downstream (see Fig.6 in King, 1984, part 1). However, ice particles after impact with the fuselage rebound back into the airflow. Ice particles, after the first rebound, may experience multiple bouncing. This phenomenon was observed in wind tunnels and is well reproduced in CFD simulations (e.g. Korolev et al JTECH, 2013). One of the consequences of this effect is an enhanced concentration of ice particles around the fuselage including side and bottom locations. This is results in a principal difference compared to the King's (part 1 and 2) work, which was focused on the trajectories of liquid droplets. In this regard, it is important to consider the enhancement of ice concentration not only at the top of the fuselage, but all the way around it. This effect may equally affect IWC measurements at the side and bottom locations. This question should be properly addressed.

We address the point of ice particle bouncing in the new Section 2 (Methodology). We have not performed CFD calculations that consider bouncing, since one advantage of the approach of comparative IWC measurements is that all effects influencing IWCs, including ice crystal bouncing, are contained in the measurements. We argue that, as soon as the ice particle sampling at one or both probing positions is seriously disturbed, the IWC measurements will differ significantly from each other. Hence, a reliable agreement between IWCs from two different instruments mounted at two different positions is a reasonable indication for an applicable IWC measurement. Smaller effects influencing IWCs result in the observed scatter of the IWCs which is discussed in the new Section 5.1.2.

4. CFD simulations showed that particle trajectories are sensitive to air density air. Therefore, the dimensions of the shadow and enhancement ice particle zones depend on the air density air along with other parameters such as TAS, AoA, etc. This is a very important issue and it should be properly addressed in this study. Could you also comment on the effect of air on the dependences of IWC ratio vs Rice shown in Fig.10?

We are aware of the effect of the density of air on particle trajectories and thus the shadow and enrichment zones and note that now in Section 2 (Methodology). However, we feel that a discussion of this effect (also with respect to the IWC ratio vs Rice shown in new Fig. 12) is

beyond the scope of this study. Our main aim here is the evaluation of IWC measurements on aircraft by means of the comparative approach described now in Section 2. As outlined in the previous point (3.), one advantage of the approach of comparative IWC measurements is that all effects that seriously influence IWCs, including also the effect of air density, would appear significantly in the measurements.

5. Page 10. The equation mean mass radius Rice = IWC/Nice should be written as
                    Rice = (3IWC/4πice Nice)ˆ 1/3.
I believe this a typo. Unfortunately, no information about ice was provided in the text. Since the size-to-mass parametrizations
                    M=aRiceˆ b
was applied for the IWC calculation, then ice is a function of Rice, i.e.
                    ice = 3aRiceˆ (b-3)/4π.
Therefore, the mean mass radius should be calculated as
                    Rice = (IWC/aNice)ˆ 1/b.
Could you please clarify how Rice was calculated?

$R_{ice}$ was calculated as $\left( \frac{3 \cdot IWC}{4\pi\rho \cdot N_{ice}} \right)^{1/3}$ with $\rho = 0.92$ g/cm$^{-3}$ and we have changed the equation accordingly.

6. It is important to indicate the distance of the TWC probes inlets from the fuselage and from the nose of the airplane. This is necessary to understanding the effect of the probe's location on the accuracy of its measurements. Along this way, it would be beneficial to include a summary table with the positioning of the TWC probes, type of the airplane, name of the project, TWC probe, particle probe used as a reference, etc.

We have included a table containing all necessary information in the manuscript (Table 1).

7. The diagrams in Figs 7, 8, 9 in their present form visualize the scattering of the IWC points. However, it is difficult to judge about the biases and the degree of scattering of the data points. It is suggested to add a linear regression line, indicate a relevant linear equation, standard deviation, and correlation coefficient in each diagram. This information will help to quantify of the degree of agreement between the IWC measurements. Please also provide the averaging time used for the data these diagrams.

We have calculated IWC-IWC linear regression and correlation coefficients, except for the Roof/Wing measurements on HALO (new Figure 10), since there is no correlation between the roof and wing IWCs. We have indicated the correlation coefficients in the plots and the coefficients in the figure captions. However, we have not plotted the regression lines into the graphs, because the graph will beome visually too confusing together with the frequencies and 1:1, 1:10 and 1:2.5 lines..
The time resolution of the measurements is now stated together with the instrument descriptions in Sections 4.1.2 and 4.2.

8. The IWC calculated from the cloud particle probes (CAS-DPOL, CIP-G,2DS) was used as a reference for the TWC probes (FISH, HAI, Waran) measurements.

The IWC from the cloud spectrometers is not used as a reference, our approach is to evaluate the IWC measurements by a comparative assessment, as now described in the new Section 2.

The processing of the scattering and imaging probes are sensitive to the algorithms and assumptions employed in the processing software. Thus, CAS-DPOL is usually calibrated in assumption that the cloud particles are spherical water droplets. Were any corrections for ice applied for the CAS-DPOL data?

We have calculated the size bins of the CAS-DPOL under the assumption of aspherical particles and found the expected differences. However, after merging the new bins for Mie ambiguities, the differences between the bin sizes between spherical and aspherical particles was so small that we decided to use only one set of size bins. This is described in Meyer et al. (2012).

What algorithms and corrections were used during the processing of the 2D probe's data?

We have used an inter arrival time algorithm to account for particle shattering and also rejection schemes for out of focus, end diode and streaker particles. This is described in detail in Meyer et al. (2012) and Luebke et al. (2016). We have referenced these papers now in the new Section 4.2.

What are the typical, min, and max number of particles in the CAS, CIP and 2DS data? Please provide an assessment of the statistical significance of PSDs used for the IWC calculations. Statistically insignificant PSDs may result in large random errors in IWC calculations. These questions should be elaborated upon and explained in the text. The assessment of the errors in the IWC calculations for the particle probe data should be provided as well.

Good point, this might be another source contributing to the scatter of IWC measurements, because due to the nature of cirrus clouds - thin, cirrous - their particle statistics is never satisfying. We mentioned that now in the new Section 2.
To achieve a better $PSD_{ice}$ statistics, a much larger aircraft instrument sampling volume would be needed, which is beyond current technology. The other way to enhance the particle counts would be to chose longer averaging times. However, then the already low resolution in space is further reduced and cloud free areas might be assigned to clouds - we decided here to keep the high time/space resolution and accept a reduced statistical significance. We don't know the exact values the referee asks for, but we provided the total number of IWC data points in new Figs. 9, 10 and 11, they range between about 7000 and 54000 data points.
Nevertheless, a reasonable indication that large random errors in IWC caused by bad counting statistics of the particles does not greatly influence the IWC derived from the $PSD_{ice}$ is again the scatter of IWCs observed Figs. 9 and 11 with most of the points close to the 1:1 line.

9. The diagrams in Fig.10 are supportive of the statement about oversampling of small particles and undersampling of large particles at the roof location. Similar diagrams should be provided for the side and bottom locations of the TWC inlets on Geophysica and WB57. Otherwise, one could argue that the 'duck' type behavior of the IWC ration vs Rice is a result of the errors in calculations of IWC from the particle probes.

With the more detailed explanation of the methodology (Section 2) we used here, we hope it is clear now that the 'duck' type behavior of the ratio IWC/$R_{ice}$ is not caused by errors in calculations of IWC from the particle probes - if that would be the case then an agreement of IWC measurements (as shown in new Fig. 9) would not be possible.

Since we have already added three Figures to the manuscript (15 Figures altogether now) we decided not to provide these additional plots to again confirm our findings. Another reason is that a similar Figure cannot be produced for MacPex, because the total number of ice crystals starts is only available for crystals larger than 15 $\mu$m (instead of 3 $\mu$m), thus the mean mass size cannot be calculated.

10. Page 4, Line 15: 'However, isokinetic sampling (= the flow inside the inlet is the same as in the free flow), which in principle enables the undistorted measurement of H2Otot, is not possible for fast flying aircraft, since the air flow speed is always much higher than the velocity inside of the inlet.' The airborne version of the isokinetic probe for measurements of cloud condensed water was designed by NRC: (Davison, C., J. MacLeod, J. Strapp, and D. Buttsworth, 2008: Isokinetic total water content probe in a naturally aspirating configuration: Initial aerodynamic design and testing. Proc. 46th AIAA Aerospace Sciences Meeting and Exhibit, Reno, NV, American Institute of Aeronautics and Astronautics, AIAA 2008-435. [Available online at http://arc.aiaa.org/doi/abs/10.2514/6.2008-435.]) This probe was successfully operated during several field campaigns on different aircrafts. Some results were published in JTECH.

The sentence is changed to 'However, as explained in the following, a deviation of the gas streamlines is desirable when sampling cirrus clouds, since cirrus are very thin and their IWC correspondingly small. To this end, so called 'nearly virtual impactors' (see Figure 2) are used for the collection of cirrus ice particles. ...'

11. Traditionally, condensed water content is measured in g/m3 (liquid, ice or total water content) or g/kg (mixing ratio). These units are well adapted by the cloud and climate modeling communities (both g/m3 and g/kg), remote sensing community (g/m3), aviation industry (mainly g/m3). The present paper is utilizing non-conventional units in the cloud physics community (ppmv) in order to describe condensed water content. This unit is usually used to describe concentration of a gas phase, rather than to characterize the weight of a liquid or solid phase per unit volume. This unit is mainly employed by the subcommunity formed around the evaporators used for measurements of the condensed cloud phase (e.g. FISH, HAI, Waran, etc.). I am not sure that employing this unit adds clarity; rather, it creates barriers in the dissemination of the IWC measurements that employ this unit. In my opinion, the cloud and climate modeling communities and the remote sensing community are unlikely to switch to this unit. The aviation industry is quite conservative, and it most likely they will ignore the measurements of condensed water content in this unit. I recommend using the conventional units of g/kg or g/m3. At minimum, I suggest using additional axes with conventional units in Fig. 7, 8 ,9, 10.

We are aware that traditionally g/m$^{-3}$ is used as unit for condensed water and account for that in new Figure 3, where we show IWC vs. temperature in both units, ppmv (volume mixing ratio) as well as mg/m$^{-3}$. The reason that we prefer volume mixing ratios is that it is a conserved quantity, i.e. they do not change with temperature and pressure and are therefore better

comparable with each other. In our publications, we use to show a graph with two panels, one for each unit, here Figure 3.

Minor comments:

1. Page 2, Line 11: 'The IWC of a cirrus is a bulk quantity which is composed of the sum of all ice particles. . .' The term 'of a cirrus' is redundant here. This statement is relevant to any cloud, not just cirrus.

Changed.

2. Page 2, Line 11: It should be '. . .the sum of all ice particles masses. . .'

Changed.

3. Page 2, Line 15: 'In particular, King (1984) shows that above the roof of an aircraft the sampling of particles is disturbed.' Strictly speaking, the sampling of particles is disturbed everywhere around the fuselage. However, the scale of this disturbance is different. Please reword this sentence.

Changed.

4. Page 2, Line 16: 'However, to simulate and quantify losses or enrichment of ice particles and the effect on IWC at a specific position of an aircraft is hardly possible, since this depends on the prevailing particle size distribution and also the irregular shape of the ice crystals.' This is a too strong of a statement. The irregular ice particle shapes can be replaced with spheres with equivalent aerodynamic size. For example, particle trajectory analysis can be performed using spheres with the mass density calculated from size-to-mass parametrizations M=aDˆ b.

The sentence is changed to 'However, to simulate and quantify losses or enrichment of ice particles and the effect on IWC at a specific position of an aircraft is hardly possible, since this depends on the prevailing ice particle size distribution and also th flight conditions.'

5. Page 2, Line 27: 'The IWC of cirrus can be recorded from aircraft either by bulk cloud measurements using airborne closed path hygrometers mounted behind an inlet tube or via integration of the ice particle number size distributions (PSDice) measured by cloud spectrometers. In both cases, the ice particles must be properly sampled before the measurement.' Hot-wire probes are missed in this statement.

We did not mention hot-wire probes because this technique is not to be recommended for ice crystals because the crystals bounce from the wire.

6. Page 2, Line 29: 'The bulk IWC is less error prone in comparison to the IWC from PSDice in case of an undisturbed measurement.' This is a questionable statement. Both techniques have its own problems and advantages.

The sentence in changed to 'The bulk IWC is less error prone in comparison to the IWC from PSDice in case of undisturbed ice particle sampling.'

7. Page 3, Line 1: replace 'Fore' to 'For'.

Changed

8. Page 3, Line 18: 'To precisely detect H2Otot' replace by 'To precisely measure H2Otot'

We have changed the sentence to 'To precisely determine H2Otot' to avoid to use 'measure' too often.

9. Page 4, Line 6: 'To specify the size ranges of the 'smaller' and 'larger' cloud particles, CFD calculations for the specific conditions of fuselage shape, aircraft speed and inlet distance from the nose of the aircraft need to be performed.' This sentence is disconnected from the following text and it appears to be redundant.

The sentence is changed to 'To specify the aforementioned size ranges of the 'smaller' and 'larger' cloud particles, CFD calculations for the specific conditions of fuselage shape, aircraft speed and inlet distance from the nose of the aircraft need to be performed.' to make clear that it refers to the previous text.

10. Page 4, Line 7: 'Very roughly, cloud particles with radii $<30\mu$m can be assumed to belong to the smaller, while those $>30\mu$m are associated to the larger part of the cloud particle size spectrum at jet aircraft with high air speeds." What is the basis for this statement? References should be provided here.

The basis for this statement is this study, thus there is no reference.

11. Page 4, Line 23: ". . .shattering into small artifacts at the cloud probes head. . ." should be ". . .shattering into small fragments at the cloud probes' housing. . ."

The sentence is changed to 'Ice crystal shattering into small fragments at the cloud probes head ...'

12. Page 4, Line 23: "However, for the calculation of the IWC, the uncertainty from shattering does not play a significant role since the shattered crystals still contribute to the integrated mass of PSDice." This sentence should be reworded.

The sentence is changed to 'However, it (shattering) does not play a significant role for the calculation of the IWC, since the ice fragments contribute to the integrated mass of $PSD_{ice}$ in the same way as the original large crystal.

13. Page 4, Line 9: IWCS should be IWCs

There is no 'IWCs' at this place, I guess you mean Page 5, Line 10 $\rightarrow$ changed.

14. Figure 11. The y-labels are not easily legible. Please enlarge the font size.

Changed.

---

## Author Response (AR2)

Answer to the second comments of the reviewers

**Ice particle sampling from aircraft –**
**influence of the probing position on the ice water content**

by

Afchine, A., Rolf, C., Costa, A., Spelten, N., Riese, M., Buchholz, B., Ebert, V., Heller, R., Kaufmann, S., Minikin, A., Voigt, C., Zöger, M., Smith, J., Lawson, P., Lykov, A., Khaykin, S., and Krämer, M.

*Correspondence to:* Martina Krämer (m.kraemer@fz-juelich.de)

Many thanks again for taking the time to read and comment our manuscript. We hope that the new version is satisfactory. Our answers to all points are written in blue. A manuscript with tracked changes is at the end of this document.

Report #2 Submitted on 09 Apr 2018 Anonymous Referee #1

Review of "Ice particle sampling from aircraft-Influence of the probing position on the ice water content." By Afchine et al.

Recommendation: Requires revision for publication

The authors have made a number of changes in the revised manuscript, and added some CFD simulations, all of which enhance the quality of the manuscript. However, I still think that the authors are not adequately acknowledging the uncertainties associated with their results for the reasons stated below:

1) The authors categorically state that "the wing IWC is derived from the measurements of PSD_Ice, that should be only weakly influenced by flow perturbation effects." Although it is believed that this is the case, the quality of the wing IWC will depend on exactly where the wing probes are mounted (for mounting positions see Table 1). Typically these probes are mounted as far beneath and ahead of the leading edge of the wing as possible, and there may be enhancements or shadowing of concentrations depending on the exact mounting location. Is this something the CFD simulations can address?

Yes, this can be addressed by CFD calculations. For the HALO aircraft, comprehensive CFD studies had been performed by the DLR flight facilities during the modification of the plane to a research aircraft to determine the optimal position for particle sampling. However, we are not allowed to show these calculations.

In Section 4.2 we now write (new text in italics):
'The mounting positions *(distance from leading edge of the wing and distance to wing surface)* are listed in Table 1. *Comprehensive CFD studies had been performed during the modification of the plane to a research aircraft to determine the optimal position for particle sampling (but without permission to be shown).*

This uncertainty in the wing IWCs must be better acknowledged (see answer to point 2)) rather than assuming that the wing IWC is a reference value.

It seems that there is a misunderstanding: we do not assume that the wing IWC is a reference value. In Section 2 (Methodology) we explained that

'.. for the evaluation of the quality of IWC measurements an experimental comparative approach of IWC measurements is useful.'

'.. a reliable agreement between IWCs from two different instruments mounted at two different positions is a reasonable indication for an applicable IWC measurement.'

'.. the IWC deviations from each other can be quantified by using the comparative IWC approach.'

Further, their statement that "most favorable for an undisturbed sampling on aircraft is most likely the position under an aircraft wing" is not necessarily true as many aircraft mount the probes so that the probes are positioned ahead of the aircraft wing to get a more clear flow.

In Section 2 Methodology we have changed (new text in italics):

'IWC measurement at a differing position, here at the aircraft wing, which is least susceptible to flow disturbances *if it is properly positioned (see Section 4.2).* In this study, the wing IWC is derived from the measurements of PSDice *(see also Section 4.2)*, that should be only weakly influenced by flow perturbation effects.'

The authors seems to acknowledge this to some extent later on when they state that "they are in most cases mounted below the aircraft wings with sufficient distance to the wing and aircraft body", but they do not define what sufficient is.

We apologize for another confusion - when we say 'position under an aircraft wing' we meant that the probes heads are positioned ahead of the aircraft wing and close to the planes body. The 'distance from leading edge of the wings' and 'distance to wing surface' are listed in Table 1.

In Section 3.1.1 we have changed (new text in italics):

'Most favorable for an undisturbed sampling on aircraft is most likely the position under an aircraft wing *with the probes head ahead of the aircraft wing...*'

In Section 3.2 we have changed

'They are in most cases mounted below the the aircraft wings with distances ahead of the wing and from the aircraft body to minimize particle losses or enrichment due to distorted cloud particle trajectories ...'

In Section 4.2 we now write (new text in italics):

'The mounting positions *(distance from leading edge of the wing and distance to wing surface)* are listed in Table 1. *Comprehensive CFD studies had been performed during the modification of the plane to a research aircraft to determine the optimal position for particle sampling (but without permission to be shown).*

2) Perhaps the greater uncertainty the calculation of the wing IWC is the use of an M-D relation to derive the IWC from a measured size distribution. This manuscript does not adequately acknowledge the uncertainty in the calculation of the IWC from a size distribution. The authors state that "a reliable agreement between IWCs from two different instruments mounted at two different positions is a reasonable indication for an applicable IWC measurement." But, if the basis of the choice of m-D coefficients is to get better agreement with the IWCs made in another location, one is merely obtaining what one assumed in the derivation of the m-D relations. Thus, it is not surprising that the IWC_PSD agrees with the FISH IWC given that this agreement was used to justify the choice of m-D relations used: this is a bit of circular logic. This point is also made on page 13 where it states that "the good agreement between the two measurements . . . shows the validity of the m-D relation used to calculate the IWC from the PSD."

We did not choose the m-D relation so that the IWC from FISH fits best to the NIXE PSD-IWC. The first deployment of both instruments was on HALO 2014, where we found the deviations between the IWCs (Figure 10) At that time we had chosen a m-D relation based on Mitchell et al. (201) (see also Krämer et al., 2016 and Luebke et al., 2016).

We were very happy when we found the agreement between FISH-IWC and NIXE PSD-IWC on Geophysica in 2017 - that motivated us to write this paper. Let me note again that our intention is not to use the PSD-IWC as a reference, but to compare IWCs from two instruments and get insights on the influence of the sampling position from the differences.

I also have to make a comment on Figure 8, which the authors use to state that the IWCs derived from PSDs are not very sensitive to the choice of m-D relation: note that 8 orders of magnitude are included on the vertical axis so visually there appears to be little dependence. However, some of the differences are not that small. The writing and analysis needs to be more quantitative so that it can be determined what a large and small difference is.

[Figure]

The sensitivity of the IWCs derived from PSDs to the m-D relation is now also shown in Figure 8, right panel (see plot on the left), where we plotted IWCs calculated from the 10 different m-D relations vs. their mean IWC for one flight during ML-CIRRUS. It can be seen from the Figure that the IWCs from the m-D relations are at most around the factor 1.5 over the entire IWC range. Specifically, 55% of the data range

between 1:±1.2, while 19/26% can be found in the ranges 1:-(1.2 to 1.5) / 1:(1.2-1.5). This is now noted in addition in Section 4.2 and also in Section 6 (Summary and conclusions).

3) I don't think the authors accurately describe how ice crystal bouncing might be affecting the calculation of the IWC. The authors state that shattering "does not play a significant role for the calculation of the IWC since the ice fragments contribute to the integrated mass of PSD_ice in the same way as the original large crystal." However, this statement is not true. The shattered artifacts are typically generated from large ice crystals hitting the tips of the probes which are outside of the sample volume, with the small remnants then being swept into the sample volume. I agree that given the use of the Korolev tips, the impact of the shatter artifacts on IWC is most likely minimal.

Bouncing of ice crystals is mentioned in Section 2: 'Further, bouncing ice crystals may break and the small fragments may enter the IWC sampling areas ... '

In Section 3.2, we have now added (new text in italics):

'Ice crystal shattering into small fragments at the cloud probes head is a source of error in PSD_ice.However, this does not play a significant role for the calculation of the IWC ,– *for cloud probes equipped with anti-shattering inlet tips –* since the ice fragments contribute to the integrated mass of PSDice in the same way as the original large crystal. *For those cloud spectrometers that use anti-shattering tips and data evaluation algorithms, ice fragments from large shattered ice crystals can be considered (Korolev et al., 2011). However, without these tools ice crystals from outside could shatter at the inlet tips and the small fragments are then being swept into the sample volume.*'

Despite these limitations, I still think that the manuscript makes a contribution to our scientific understanding of probe position on measured IWC. However, I think the uncertainties in the work need to be better acknowledged. Further, if a more quantitative analysis of the uncertainties could be made the manuscript would read better. Further, use of terms like "significant", "small" and "large" should be avoided and replaced with more quantitative explanations of what the differences are.

We thank the referee for the positive assessment of the scientific message of the manuscript; we think that with the quantification of the dependence of the wing IWC on the m-D relation (Section 4.2) together with the Sections 'Methodology' (2.) and 'Scatter of IWC measurements' (5.1.4) the sources and extent of uncertainties are now being thoroughly discussed. We also have checked the manuscript for the terms "significant", "small" and "large" and changed them wherever possible.

Report #1 Submitted on 03 Apr 2018 Anonymous Referee #3

Evaluation of the revised manuscript titled: "Ice particle sampling from aircraft – influence of the probing position on the ice water content" by Afchine et al.

The revised manuscript was improved and addressed many questions. However, these clarifications brought up other serious issues, which were hard to identify in the original submission.

1. The authors evaluate comparisons between FISH/HAI/WARAN probes (Fig.9) based on the visual assessment of proximity of the scatter points to the 1:1 line in the log-log coordinates. This assessment led to the conclusions in sections 5.1.1 and 5.1.4, like "Most of the measurements symmetrically spread around the 1:1 line within a factor of 2.5, which can be considered as a good agreement". However, after addressing comment #7 and qualifying the relationships between the IWCs measured by FISH, HAI and WARAN it becomes clear that these dependencies are explicitly non-linear. The FISH and HAI IWCs are related as Y =0.481X+ 0.270, and FISH and WARAN IWCs correspondingly are related as Y = 0.651X+ 0.562. Here, Y and X represent log(IWC) of relevant instruments. So, the first dependence is close to a quadratic relationship between the FISH and HAI IWCs, whereas the second one is close to 3/2 power dependence between FISH and WARAN IWCs. Such relationships between measurements of the probes mounted in the same location put the accuracy of their IWC measurements into question.

We apologize that we inaccurately described the way we did the regression calculations which has led to confusion. What we wanted to express with Y,X= log(IWC) is that we performed the regression calculations over the logarithms of IWC. The reason is that using logarithmic variables can reduce their range and negative effects of outliers can be decreased. This method has also been used by Davis et al. (2007) when comapring IWCs from different instruments. Of course, the IWCs of the regression lines are calculated using X, Y = IWC. We have changed the text in the manuscript accordingly.

To show that the relation between the IWCs are indeed linear, we have drawn the new regression lines in green in the below Figure (Figure 9, middle and right panel). The new regressions are explained in the answer to your point 3.

[Figure]

2. The response to the comment #9 includes the statement "With the more detailed explanation of the methodology (Section 2) we used here, we hope it is clear now that the 'duck' type behavior of the ratio IWC/Rice is not caused by errors in calculations of IWC from the particle probes - if that would be the case then an agreement of IWC measurements (as shown in new Fig. 9) would not be possible." As follows from the regression equations in the figure caption to Fig.9 there is no such agreement. The relationships between the three IWC probes appear to be non-linear. This fact may put into question the results presented in Fig.12.

See answer to point 1.

3. The absence of a correlation between the roof FISH/HAI/WARAN and wing NIXE-CAPS IWC measurements in Fig.10 requires an explanation. Otherwise, one may doubt the performance and operability of these instruments during the ML-CIRRUS campaign. Since this is directly related to the objective of this study, this question should be properly explored and addressed, rather than just stating the absence of correlation in section 5.1.2 as a matter of fact.

The explanation are the inlet positions of FISH/HAI/WARAN at the roof. To make that clear, we have added at the end of Section 5.1.2:

'No correlations between the IWC measurements can be observed here, as expected when sampling ice crystals on the roof of an airplane, where the measurement is influenced by shadow/enrichment zones for larger/smaller particles (see Section 3.1.1). The structures of the IWC deviations seen in Figure 10 will be further analyzed in Section 5.2. seen in Figure 10 will be further analyzed in Section 5.2). What can already be seen when comparing the scattering of IWCs with that around the 1:1 line of the NIXE cloud spectrometer IWCS (see Figure 8, IWCs from different m-D relations), is that the m-D relation is not the cause for the deviations seen in Figure 10.

The correlation coefficient between FISH and HAI (i.e. $R^2$=0.33) seems to be overly low too. This fact also requires explanation.

Thanks for the comment on the low correlation coefficient between FISH and HAI. The reason for this is the higher detetction limit of HAI in comparison to FISH (see Figure 3). We have now repeated the calculation of the regressions (except for MacPex where mostly high IWCs were measured) by taking into account the lower detection limits of the instruments. As a result, the FISH/HAI regression coefficient is now $R^2$=0.82 . We have changed the text in the manuscript accordingly.

[revised manuscript text omitted]